# HIV-1 Drug Resistance Detected by Next-Generation Sequencing among ART-Naïve Individuals: A Systematic Review and Meta-Analysis

**DOI:** 10.3390/v16020239

**Published:** 2024-02-02

**Authors:** Fei Ouyang, Defu Yuan, Wenjing Zhai, Shanshan Liu, Ying Zhou, Haitao Yang

**Affiliations:** 1Key Laboratory of Environmental Medicine Engineering of Ministry of Education, Department of Epidemiology and Health Statistics, School of Public Health, Southeast University, Nanjing 210009, China; 220213961@seu.edu.cn (F.O.); 230239083@seu.edu.cn (D.Y.); 220223638@seu.edu.cn (W.Z.); 220223703@seu.edu.cn (S.L.); 2Department of HIV/STD Control and Prevention, Jiangsu Provincial Center for Disease Control and Prevention, Nanjing 210009, China; 3Jiangsu Health Development Research Center, Nanjing 210029, China

**Keywords:** HIV, next-generation sequencing, pretreatment drug resistance, ART-naïve, antiretroviral therapy, meta-analysis

## Abstract

Background: There are an increasing number of articles focused on the prevalence and clinical impact of pretreatment HIV drug resistance (PDR) detected by Sanger sequencing (SGS). PDR may contribute to the increased likelihood of virologic failure and the emergence of new resistance mutations. As SGS is gradually replaced by next-generation sequencing (NGS), it is necessary to assess the levels of PDR using NGS in ART-naïve patients systematically. NGS can detect the viral variants (low-abundance drug-resistant HIV-1 variants (LA-DRVs)) of virus quasi-species at levels below 20% that SGS may fail to detect. NGS has the potential to optimize current HIV drug resistance surveillance methods and inform future research directions. As the NGS technique has high sensitivity, it is highly likely that the level of pretreatment resistance would be underestimated using conventional techniques. Methods: For the systematic review and meta-analysis, we searched for original studies published in PubMed, Web of Science, Scopus, and Embase before 30 March 2023 that focused exclusively on the application of NGS in the detection of HIV drug resistance. Pooled prevalence estimates were calculated using a random effects model using the ‘meta’ package in R (version 4.2.3). We described drug resistance detected at five thresholds (>1%, 2%, 5%, 10%, and 20% of virus quasi-species). Chi-squared tests were used to analyze differences between the overall prevalence of PDR reported by SGS and NGS. Results: A total of 39 eligible studies were selected. The studies included a total of 15,242 ART-naïve individuals living with HIV. The prevalence of PDR was inversely correlated with the mutation detection threshold. The overall prevalence of PDR was 29.74% at the 1% threshold, 22.43% at the 2% threshold, 15.47% at the 5% threshold, 12.95% at the 10% threshold, and 11.08% at the 20% threshold. The prevalence of PDR to INSTIs was 1.22% (95%CI: 0.58–2.57), which is the lowest among the values for all antiretroviral drugs. The prevalence of LA-DRVs was 9.45%. At the 2% and 20% detection threshold, the prevalence of PDR was 22.43% and 11.08%, respectively. Resistance to PIs and INSTIs increased 5.52-fold and 7.08-fold, respectively, in those with a PDR threshold of 2% compared with those with PDR at 20%. However, resistance to NRTIs and NNRTIs increased 2.50-fold and 2.37-fold, respectively. There was a significant difference between the 2% and 5% threshold for detecting HIV drug resistance. There was no statistically significant difference between the results reported by SGS and NGS when using the 20% threshold for reporting resistance mutations. Conclusion: In this study, we found that next-generation sequencing facilitates a more sensitive detection of HIV-1 drug resistance than SGS. The high prevalence of PDR emphasizes the importance of baseline resistance and assessing the threshold for optimal clinical detection using NGS.

## 1. Introduction

Effective antiretroviral therapy (ART) has significantly reduced HIV-related morbidity and mortality among people living with HIV [1,2]. Unfortunately, increasing evidence clearly indicates that HIV drug resistance (HIV-DR) has developed into a realistic barrier to the effectiveness and long-term sustainability of antiretroviral therapy (ART) [3,4]. The primary causes for HIV-DR include the effectiveness of the antiretroviral regimen and patient variables related to the establishment and dissemination of HIV drug resistance mutations, such as insufficient adherence to ART, limited access to medications, drug toxicity and drug–drug interactions, and gaps in HIV preventive measures, care, and treatment cascades [5].

PDR refers to resistance detected among individuals who have not yet initiated ART, or who have prior antiretroviral drug (ARV) exposure and are re-initiating first-line treatment. PDR may be either acquired through earlier antiretroviral medication (ARV) exposure or prior exposure to ARVs, inducing a continuous public health issue that could have an impact on the use of ART among people living with HIV [6].

According to the WHO, nonnucleoside reverse transcriptase inhibitor (NNRTI)-based ART regimens are recommended as a first-line treatment regimen in most countries [7]. The prevalence of PDR to NNRTIs has increased to over 10% with the widespread application of ART in low- and middle-income countries [7]. Based on this status, dolutegravir (DTG) has been recommended by the WHO as the preferred first- and second-line treatment for all population groups since 2019 because it has fewer side effects than NNRTIs and a high genetic barrier for developing drug resistance as an integrase strand transfer inhibitor (INSTI) [7,8]. PDR can restrict ARV options and compromise the effectiveness of ART [9]. People living with HIV have to purchase other ARVs due to the high prevalence of PDR to widespread first-line treatment regimens, which increases their economic burden. The growth of mortality and an increasing number of newly infected HIV individuals due to bad ART outcomes would hinder the global prevention and control of the HIV/AIDS epidemic [9,10]. If the problem of HIV-1 drug resistance cannot be effectively controlled, it may lead to an increase in drug-resistant viruses and the emergence of multidrug resistance [11]. Thus, it is important to conduct continuous resistance surveillance among people living with HIV. However, due to high costs, resistance testing is not feasible in the majority of low- and middle-income nations.

The selection of ART is mostly dependent on an HIV genotypic drug resistance test. Population-based Sanger sequencing has been a standard procedure for years to detect all HIV resistance mutations. However, Sanger sequencing, due to its lower sensitivity, generally fails to detect low-abundance drug-resistant HIV-1 variants (LA-DRVs) present at frequencies below 15–20% within virus quasi-species [12,13,14]. LA-DRVs may be present in HIV-infected individuals before treatment initiation but may be screened out during antiretroviral therapy, ultimately leading to treatment failure. Therefore, monitoring these mutations is critical to achieving effective HIV therapy.

Next-generation sequencing (NGS) is capable of detecting both high- and low-abundance DRVs and has proven suitable for identifying HIV-1 drug resistance mutations [15,16]. The use of NGS techniques in genotypic resistance testing holds the promise of improved HIV diagnosis and surveillance at a lower cost and level of automation. Using a lower threshold than the 20% conventionally utilized for SGS can improve the sensitivity in identifying patients at risk of virological failure during first-line NNRTI-based antiretroviral treatment [17]. LA-DRVs may be significantly associated with a dose-dependent increased risk of virologic failure of first-line ART [18,19]. Other studies have shown that the minority that were resistant are not associated with the failure of antiretroviral therapy without prior exposure to ARV [20,21,22,23].

Given the diversity of study designs and laboratory NGS techniques employed, the optimal detection threshold for clinical relevance has not yet been determined. A 2% threshold is more stable and more likely to be selected as the reporting threshold, whereas a 1% threshold tends to introduce artifactual errors [24,25,26]. Other researchers have suggested that lowering the detection threshold for pretreatment resistance to 5% can improve the ability to identify patients with virologic failure compared to the traditional 20% threshold [6,17]. The 20% threshold, on the other hand, is a standard reference point for defining predominant resistance mutations with established clinical significance. These detection thresholds are important cut-off points that can guide therapeutic decisions in HIV management. Therefore, our study focuses on the prevalence of PDR and the pooling of drug-resistant mutations using NGS at five thresholds (1%, 2%, 5%, 10% and 20%). We hope to understand the baseline drug resistance by summarizing the results of related studies and further analyzing them according to different detection thresholds and detection techniques in antiretroviral drug-naive individuals. Meanwhile, we will also summarize the mutations at different thresholds. We aspire to gain a more comprehensive understanding of drug resistance along with establishing further knowledge in implementing ultrasensitive HIV-DR surveillance in routine assays.

## 2. Materials and Methods

### 2.1. Search Strategy and Selection Criteria

In this study, we searched the PubMed, Web of Science, Scopus, and Embase databases, and the relevant literature was supplemented by a manual search up to 30 March 2023. The search was conducted using MeSH terms and keywords linked to HIV, AIDS, Human Immunodeficiency Virus, Acquired Immune Deficiency Syndrome, drug resistance, mutation, high-throughput, next-generation sequencing, deep sequencing, ultra-deep sequencing, high-throughput nucleotide sequencing, and pretreatment. The reference lists of included studies were checked to find additional relevant articles. The systematic review protocol (registration number CRD42023448773) was submitted to PROSPERO.

### 2.2. Study Inclusion and Exclusion Criteria

The retrieved articles were screened according to the following inclusion and exclusion criteria. Inclusion criteria were as follows: (1) the study participants were people living with HIV with no prior history of ARV exposure as our aim was to specifically assess baseline resistance prior to initiating any treatment; (2) drug resistance mutations were reported by applying NGS or deep sequencing technologies; (3) the number of included reports and the number or proportion of drug resistance mutations were clearly reported; (4) HIV gene sequences were submitted to the Stanford University Drug Resistance Database or the WHO recommended drug resistance mutation list; and (5) the full text was accessible.

Exclusion criteria were as follows: (1) reviews (narrative or systematic), meta-analyses, letters to the editor, case series or case reports, or pooled analyses of raw data (they provided very limited information or the full-text information was duplicated); (2) studies in which genotypic resistance testing was conducted using allele-specific polymerase chain reaction or oligonucleotide ligation assay methods; (3) studies examining only specific mutations; (4) research on HIV co-infection and HIV-2; and (5) studies with fewer than 10 participants and those published using the same data. Studies with small sample sizes may not accurately reflect the prevalence and patterns of resistance due to limited statistical power.

### 2.3. Data Extraction

The retrieved literature was progressively screened using Endnote, where two researchers independently screened the titles and abstracts, and they assessed and recorded the full text. The two researchers extracted the data according to a predetermined data extraction form. After data extraction, the results were compared and validated by the two researchers. When disagreements existed, they were resolved by consensus or arbitration by a third researcher. We extracted the following data from each study: first author, year of publication, sample source, sampling time, number of successfully amplified sequences, and number of drug resistance cases at five thresholds (1%, 2%, 5%, 10%, and 20%). The detection threshold in the context of HIV NGS refers to the minimum proportion or frequency of a variant (such as a mutation associated with drug resistance) in a viral population that the sequencing method can reliably detect. The specific threshold value can differ based on the NGS platform used, the quality of the sample, the depth of sequencing, and the data analysis pipeline. When both SGS and NGS were applied in the same studies, the number of participants with drug resistance based on the sequencing techniques was extracted independently. HIV drug resistance detected by SGS was attributed to the 20% threshold. When the studies used different algorithms to interpret resistance mutations, such as the Stanford HIV Drug Resistance Database (HIVdb), Rega Algorithm and ANRS Algorithm, we extracted the results reported using the HIVdb.

### 2.4. Quality Assessment

The JBI’s critical appraisal tools for prevalence studies (available at https://jbi.gobal/critical-appraisal-tools, accessed on 12 July 2023) were suggested for evaluating the risk of bias [27]. We assessed the risk of bias through a detailed analysis of several study parameters. These parameters included the sample’s representativeness, the sampling frame, the sampling technique, response bias, the use of proxies, clarity of case definition, the measurement’s accuracy, the uniformity of the data collection, the period of prevalence, and the suitability of the numerator and denominator. In the research context of HIV genotype resistance testing, ‘uniform data collection’ refers to the practice where all biological samples from participants, such as blood or other biospecimens, are collected under similar conditions throughout the study. For instance, samples might be obtained prior to the initiation of treatment, with consistent sample processing and storage conditions maintained. Furthermore, identical laboratory techniques and equipment are employed to perform resistance genotype assays. Interpretation and reporting of the test results adhere to uniform standards and guidelines, such as the application of consistent criteria for the identification of resistance mutations and the use of standardized interpretation algorithms. The answers were divided into “Yes”, “No”, “Unclear”, and “Not applicable”. Based on an evaluation of these items, the cut-off for research publications to be included was a JBI score of 50%, based on established practices and consensus within the research community [28]. Specifically, a study was included for analysis only if it met at least half of the quality criteria, suggesting a satisfactory bias risk level. This cut-off was employed to categorize the overall quality of evidence, with the understanding that it represents a guideline rather than an absolute standard.

### 2.5. Data Analysis

Excel was used to create data information extraction tables to organize the data. For our meta-analytic procedures, we utilized the meta package in R (version 4.2.3) (R Core Team, 2023) [29]. The meta package evaluated the combined effect value of the PDR rate among ART-naïve patients. Effect estimates were reported with 95% confidence intervals. We performed a Logit transformation on the extracted data so that the data satisfied a normal distribution. Heterogeneity was analyzed using Cochran’s Q test and I^2^ statistic. When I^2^ < 25%, 25–50% and >50%, the heterogeneity was low, medium, and high, respectively. According to the results of heterogeneity, the random effects model was chosen to estimate the overall prevalence and each ARV class of PDR at five thresholds (>1%, 2%, 5%, 10%, and 20% of virus quasi-species). Chi-squared tests were used to analyze differences between the overall prevalence of PDR using SGS and that using NGS. Chi-squared tests were used to analyze differences in the prevalence of PDR at a 2%, 5%, and 10% threshold. We crudely pooled the numbers of individuals with any mutations and those with resistance mutations and then calculated the proportion with drug resistance mutations [30,31]. Publication bias analysis was performed using Egger’s test and funnel plots, and publication bias was considered not to exist when *p* ≥ 0.05 [32]. Leave-one-study-out sensitivity analysis was performed to determine the stability of the results [33].

## 3. Results

### 3.1. The Basic Characteristics of the Literature

The study selection procedure is illustrated in Figure 1. The literature search yielded 3170 potential records from four electronic databases based on the search strategy. Additional articles were identified by experts in the field. We screened out 1330 studies after removing duplicates, and 140 articles were downloaded for full-length screening, of which 39 articles were kept [6,17,24,34,35,36,37,38,39,40,41,42,43,44,45,46,47,48,49,50,51,52,53,54,55,56,57,58,59,60,61,62,63,64,65,66,67,68,69].

Table 1 (see Appendix A for further details) lists the baseline characteristics of the literature included in this study and the results of the quality assessment. In total, these studies reported data from 15,242 participants, including data on PDR-associated mutations. The number of participants ranged from 20 to 2902, with a median of 148 (interquartile range: 60–318) individuals. Twenty-three (59.0%) studies reported sample sizes below 200. The sampling span was from 1992 to 2019. Fourteen (35.9%) of these studies aimed to investigate transmitted drug resistance, and eight (20.5%) studies aimed to investigate the presence and clinical impact of low-abundance drug-resistant variants. Thirty-one (79.4%) studies did not specify acute and chronic HIV-1 infection. Plasma was the most common specimen type (37 studies, 94.9%). The detection methods utilized with NGS were mainly Illumina NGS (22 studies, 59.0%) and 454 pyrosequencing (14, 35.9%). Twelve (30.8%) studies reported drug resistance that was detected by SGS and NGS, while the remaining studies (69.2%) reported only NGS data. In total, 13 studies (33.3%) were conducted mainly in Europe and North America, while the other (22, 56.4%) studies were conducted in low- and middle-income countries in Africa (13, 33.3%), Asia (6, 15.4%) and Latin America (3, 7.7%). All of the included publications in this study had JBI values that were over 50%.

### 3.2. The Prevalence of Pretreatment Drug Resistance at Different Sensitivity Thresholds

As Table 2 shows, we calculated the overall prevalence of PDR and that for four classes of ARVs at detection thresholds of 1%, 2%, 5%, 10%, and 20%. The random effects model yielded an average PDR of 16.69% (95%CI: 13.53–20.41%; I^2^ = 94.5%) among 30 studies. The prevalence of PDR increased with increasing detection sensitivity. The overall PDR of the 30 studies varied from 11.0% at a threshold of 20% to 29.74% at a threshold of 1%. PDR values varied less between the 2% threshold and the 10% threshold. There was no evidence of publication bias in this analysis, as indicated by the funnel plot (Appendix A) and Egger’s test (*p* = 0.936). The forest plots of the meta-analysis of PDR at the five thresholds can be found in Appendix A.

As Table 2 shows, there was an inverse correlation between the PDR rates and the detection thresholds for these four classes of ARVs. At a threshold of 20%, the PDR to NNRTIs (6.64%) was highest. Eighteen studies reported HIV drug resistance at a threshold of 1%. Due to having the highest sensitivity, the PDR at the 1% threshold was higher than the PDR at the other thresholds. At a threshold of 2%, the PDR to NRTIs (10.01%) ranked first, and the PDR to NNRTIs was 9.51%.

Comparing the 2% and 20% threshold of PDR for the four ARV classes, the ratio for the difference in NNRTIs, NRTIs, PIs, and INSTIs showed an increase of 2.37-fold (9.51%/4.01%), 2.50-fold (10.01%/4.01%), 5.52-fold (9.49%/1.72%), and 7.08-fold (1.77%/0.25%), respectively. It appeared that the PI and NRTI resistance mutations had a higher probability of being found at a more sensitive threshold.

Multiple comparisons of PDR values at different thresholds were performed. The comparisons revealed that there were significant differences between the five thresholds. At the 5%, 10%, and 20% thresholds, there were no significant differences between the three groups. When comparing the 5% and 10% thresholds (Appendix A), the prevalence of PDRs (χ^2^ = 1.89, *p* = 0.171), NNRTIs (χ^2^ = 1.59, *p* > 0.05), NRTIs (χ^2^ = 0.34, *p* > 0.05), and PIs (χ^2^ = 1.67, *p* > 0.05) did not differ significantly. However, the PDR to INSTIs was statistically different (χ^2^ = 4.62, *p* < 0.05) when comparing the PDR values at two thresholds. Compared to the other three types of ARVs, PDR was least likely to be reported again for INSTIs. There was no significant difference in the prevalence of NNRTI PDR between the 2% and 5% thresholds. Meanwhile, other prevalences of PDR showed significant differences (*p* < 0.01) (Appendix A). When comparing the 2% and 10% thresholds, a significant difference was not found for the NNRTIs (Appendix A).

### 3.3. HIV Drug Resistance Mutations of Reverse Transcriptase Inhibitors, PIs, and INSTIs among ART-Naïve Individuals

While a greater variance in DRMs was identified at thresholds as low as 1% in our dataset of 7614 patients, NNRTI resistance mutations were most frequently observed at thresholds above 20%, suggesting a higher viral prevalence of such mutations. The most common mutations with NNRTIs were V179D/E/I (7% of patients), followed by K103E/H/N/Q/R/S/T (7.2% of patients). K103N was the most common surveillance drug resistance mutation with a frequency of 3.9%. At a threshold of 20%, V179I (4.15% of patients) was the most common mutation. However, at a threshold of 1–5%, we observed that P225H (1.75% of patients) was the most common (Figure 2A).

We extracted NRTI resistance mutations from 18 studies. In total, 1495 mutations occurred in less than 20% of the viral population and 637 mutations were detected at ≥20%. The most common NRTI-related mutation was D67E/N/G (6.3% of patients), in which D67N can develop resistance to Zidovudine. The second most frequent mutation was T69N (3.6% of patients) (Figure 2B). The most common surveillance drug resistance mutations were D67E (4.0% of patients) and F77L (1.07% of patients). NRTI-associated mutations were more likely to be detected at a threshold below 20%. At a 20% threshold, S68G (1.6% of patients) was the most frequent mutation, followed by V90I (1.0% of patients).

At the 20% threshold, the most frequent PI-associated resistant mutations were A71V (5.5% of patients) and L10I (4.2% of patients). Other commonly observed PI-associated resistance mutations included A71T (4.0%), K20R (1.5%), and L10V (1.2%) (Figure 2C). M46I was the most frequent surveillance drug resistance mutation with a frequency of 1.5%, and this mutation was observed mainly as a low-abundance drug-resistant variant.

We pooled the INSRI-associated resistance mutations from 4148 participants. The most frequent INSTI-associated resistance mutations were S230N (10.5% of patients) and E138D (3.0% of the participants) at a 20% threshold (Figure 2D). However, other INSTI-associated resistance mutations were mainly minority variants which occurred between 1% and 5% and at the 20% threshold, including T66K (4.0%), Q95K (0.8%) and Q148K (0.6%). The number of participants with mutations at less than 20% and ≥20% thresholds was 687 and 683, respectively, which showed no difference. The frequencies of all drug resistance mutations included are shown in Appendix A.

In each ARV class, it was found that several surveillance drug resistance mutations tended to virtually appear, entirely or mostly, as LA-DRVs. D67EN and F77L were the most frequent NRTI-associated surveillance resistance mutations that primarily occurred as minority variants, but T215 revertants were discovered in the majority of patients. G190ASE was one of the most common minor variants of NNRTI-associated resistance mutations, while K103 predominated in the quasi-species. The same scenario appeared for the PI-associated resistance mutations: M46IL and D30N were predominately seen as LA-DRVs, whereas L90M (associated with saquinavir resistance), if present, dominated the quasi-species.

### 3.4. The Prevalence of LA-DRVs

Twelve articles reported the prevalence rate of LA-DRVs at 1–20% thresholds (Figure 3). The combined overall prevalence was 9.45% (95%CI: 5.94–14.71%; I^2^ = 90%). Sub-analysis with the ARV classes showed that the prevalence of LA-DRVs for NNRTIs was 10.4% (95%CI: 3.71–25.93%; I^2^ = 93%), for NRTIs 12.10% (95%CI: 6.52–21.36%; I^2^ = 88%), for PIs 8.86% (95%CI: 3.92–18.81%; I^2^ = 90%), and for INSTIs 4.40% (95%CI: 1.68–18.81%; I^2^ = 89%).

### 3.5. Subgroup Analysis

Data reporting and detection techniques were analyzed in subgroups. Most of the studies used Illumina NGS and 454 pyrosequencing to detect resistance mutations and resistance levels. At the 2% threshold, no study used 454 pyrosequencing to report drug resistance. Subgroup analyses showed no significant differences between the two detection methods at a threshold of 1% (*p* = 0.511), 5% (*p* = 0.601), and 20% (*p* = 0.601). Unexpectedly, the resistance mutations detected by the two assay techniques were significantly different at a threshold of 10% (*p* < 0.05) (Table 3).

Among the twelve studies that reported drug resistance using SGS, five did not report NGS drug resistance at a threshold of 20%. The results detected using both SGS and NGS were highly consistent at the 20% threshold. There was no statistical discrepancy in the prevalence of PDR to NNRTIs, NRTIs and PIs when comparing drug resistance detected with SGS at a threshold of 20%, and NGS at a threshold of 20%. When comparing drug resistance detected with SGS (20% threshold) and NGS (1% threshold), the prevalence of PDR to NNRTIs and NRTIs was significantly different (Table 4).

### 3.6. Sensitivity Analysis

Sensitivity tests were performed on the combined effect sizes of the four resistance rates in the included studies, and after deleting any of the individual studies one at a time, the combined effect values did not vary significantly, indicating that the results of the meta-analysis were stable.

## 4. Discussion

This study aimed to investigate the overall prevalence of PDR at different sensitivity thresholds using NGS and pool the drug resistance mutations by ARV class. The rate of PDR decreases with increasing detection thresholds, which means that the higher the sensitivity, the higher the level of pretreatment resistance. At a 1% threshold, the PDR rate was 1.3-fold more than the PDR at a 2% threshold. This might be because, at a 1% reporting threshold, it is more likely for errors to be introduced due to PCR amplification and different NGS data analyses than at a 2% threshold, increasing the incidence of false-positive resistance mutations [70]. Therefore, the 2% reporting threshold is often used to characterize drug-resistance mutations in people living with HIV. Tzou et al. showed a significant increase in the proportion of artificial positions of unusual mutations in samples with thresholds below 1% and lower viral loads, which may be due to PCR errors or a G-to-A hypermutation [71]. They suggested that analyzing the number of abnormal and characteristic APOBEC mutations at different NGS mutation detection thresholds may help avoid the risk of selecting too low a threshold and identifying false-positive resistance mutations.

At a 20% threshold, the prevalence of PDR was 11.08% (95%CI: 8.43–14.43%), whereas previous studies from Eastern Africa reported an overall PDR prevalence of 8.7% and 10.0% [30,72]. There was no significant difference between SGS and NGS when the 20% threshold was used to report resistance in our study. This suggests that there is good agreement between SGS and NGS when a threshold of 20% is used to report resistance mutations. NNRTIs had the highest prevalence of PDR at the 20% threshold. In contrast, when the threshold was 2%, NRTIs had the highest prevalence of PDR, followed by NNRTIs. Thus, the frequency of mutations associated with NNRTI is more likely to be greater than the 20% threshold. This might be due to the low genetic barrier (a single DNA mutation can drastically affect drug susceptibility) and prolonged marketing period of NNRTIs [73]. The most common NNRTI-associated mutations were V179I, which can largely reduce the susceptibility or virological response. NNRTI-associated resistance continues to predominate HIV-DR globally [74]. The proportion of patients with PI and NRTI LA-DRV is even higher. Zhou et al. found that the prevalence of minority drug resistance mutations was higher in ART-naïve individuals (85%; 17/20) [40]. The reason for this may be that the sample size is too small. These resistance mutations that are predominantly at a low level are preferred for impairing virus fitness and transmission. In our investigation, accessory mutations made up the majority of the PI-resistance mutations found by NGS in ARV-naïve individuals. The aggregation of these polymorphic accessory PI mutations may impair a person’s susceptibility to certain PIs such as lopinavir or nelfinavir. Therefore, a Sanger sequencing cut-off of 20%, which is weak for detecting low levels of mutations, would greatly underestimate the prevalence of PDR.

At any detection threshold, the PDR prevalence for INSTIs remains rare. Similarly, a study from Europe found INSTI-transmitted drug resistance in 0.30% of ART-naïve individuals [75], indicating that INSTI drug resistance is uncommon and unlikely to be discovered. In our study, the mutations detected by NGS were mainly discovered between 1% and 5% thresholds. The most commonly detected resistance mutation was T66K, which is associated with high-level elvitegravir resistance, while the resistance mutation detected at 20% had little impact on reducing the susceptibility to INSTIs such as S230 and E138D. Stekler et al. and Rutstein et al. showed that the major INSTI mutations were rarely found, which was consistent with our results. Given the increasing use of INSTI-based regimens in low- and middle-income countries, it is essential to perform continuous resistance surveillance of INSTIs using more sensitive technologies as well as monitoring their clinical impact [8,76]. However, the clinical relevance of LA-DRVs remains controversial [18,25].

In our study, we observed a significant difference in the detection of drug-resistant mutations when applying a 2% threshold compared to a 5% threshold. However, no statistical difference was found between the 5% and 10% threshold. Based on these findings, we assumed that the optimal threshold is more likely to be closer to 2% or 5%. The debate around the sensitivity threshold for case identification and specificity for identifying controls is still ongoing. Ávila-Ríos et al. observed that the risk of virologic non-suppression after ART initiation was associated with resistance reporting thresholds [58]. They argued that a 5% threshold might be more appropriate than a 2% threshold. The same suggestion was made by Inzaule et al., who recommended a threshold of 5% for NNRTI LA-DRV [17]. Derache et al. compared pretreatment resistance and clinical impact at 5% and 20% thresholds and showed that the risk of virologic failure was significantly higher in the presence of LA-DRVs at the 5% threshold [6]. We acknowledge that a more sensitive detection threshold can enhance an overall good diagnostic performance. However, given the diversity of study designs and the laboratory techniques employed, assessing the clinical impact of LA-DRVs on first-line ART regimens remains challenging [77,78,79,80,81]. More modeling studies are required to determine the best trade-offs and to understand the overall effect of various detection thresholds on clinical outcomes.

Our study aims to emphasize the need for continued research to provide more accurate guidance for the use of NGS technology in clinical applications. We advocate that future research should focus on conducting clinical trials to assess the impact of resistance mutations at different detection levels on treatment outcomes; establishing standardized NGS protocols to reduce variations between different laboratories; and exploring in depth the clinical significance of low-abundant resistance mutations.

This study has several strengths. First, it provides an understanding of pretreatment resistance at different thresholds by pooling and analyzing pretreatment resistance reported using NGS technology. It was found that the drug resistance profile was inversely proportional to the detection threshold. Conventional Sanger sequencing may underestimate pretreatment resistance, and more extensive pretreatment resistance monitoring is necessary. Second, in our study, we chose to exclude subjects who had received prior antiretroviral therapy (ART) in order to isolate the incidence of primary drug resistance (PDR) in a population that had not initiated treatment. Including individuals who have received prior ART may confound the results of the study because resistance patterns may be influenced not only by primary resistance mutations but also by mutations acquired during prior therapy. Due to the persistence of a potential reservoir of replication-competent HIV-1 in patients on ART, abrupt ART interruption will inevitably lead to HIV rebound and progression [82]. It has been shown that when patients harboring low-abundant resistance mutation loci stop taking their medication and then take it again, the resistant mutant strains in their bodies are rapidly selected, and thus are referred to as the dominant strain, affecting subsequent treatment outcomes [83,84].

Nonetheless, our study has several limitations. First, this study is a meta-analysis, and the data analyzed were extracted from the literature rather than using raw data, which may have led to a small bias in data selection. The variations in data across the sources are primarily attributable to methodological elements that affect the final prevalence estimate, including study design, sampling strategy, sample size, population type, survey sites, and participant selection criteria. There may be differences in the prevalence of PDR due to differences in the NGS techniques used to identify PDR in various investigations. Our study was mainly limited to the use of 454 pyrosequencing and the MiSeq platform (Illumina). The 454 pyrosequencing technology generates longer read lengths compared to the MiSeq platform, which offers higher throughput and more sequence data due to its paired-end sequencing capabilities [85]. Our study did not show a difference between the two detection technologies in identifying PDR at a threshold of 1%, 5%, and 20%. Unexpectedly, the difference found for the 10% threshold is perhaps due to the fact that Ji’s study used Tagged pooled pyrosequencing based on the 454 pyrosequencing platform, which employs sample labeling and pooling strategies to improve efficiency and reduce costs [50]. Sequences from multiple samples can be determined in a single experiment, increasing the amount of data and coverage of the experiment. Second, it is very unfortunate that we have not yet been able to identify the optimal clinical detection threshold from these data, and our results lacked a pooled analysis of the association of clinical outcomes. However, multiple previous studies have suggested that the 5% threshold may be more reproducible in the clinical setting and that resistance sites detected at the 5% mutation threshold raise the risk of virologic failure. More studies are needed to evaluate the effect of pre-existing drug resistance on treatment outcomes in patients initiating antiviral therapy. Last, we could not entirely exclude unreported prior exposure to ART even though we included papers that documented PDR in ART-naïve patients. PDR prevalence rates are often greater in those who have previously had ART compared to people who have never received ART.

To achieve the third 95% target, it is urgent to respond with appropriate measures. It is important to improve the monitoring of viral load and drug-resistant strains both before and after patient therapy. In patients with chronically low viral loads, NGS can help physicians to fully detect the occurrence of drug-resistant mutations. To provide individualized treatment, a thorough investigation of the factors causing inadequate virological suppression associated with resistance mutations prior to treatment is also required. Secondly, to improve patient adherence to treatment, HIV diagnostic and care systems still need to be continuously optimized. PI-based drug regimens or INSTI regimens can be chosen as first-line treatment regimens if the resistance rate to NNRTIs exceeds 10% [79]. In addition, the establishment of a database of drug-resistant sequences for the strains of each country is essential [86]. Collaborative efforts of microbiologists, clinicians, and bioinformaticians are needed to standardize these thresholds and harmonize them across different platforms and studies.

## 5. Conclusions

In summary, NGS has higher sensitivity and specificity in detecting low-abundance drug-resistant mutations compared to SGS, so the integration of NGS should be an important part of future HIV drug resistance research. Future research should focus on the question of the optimal threshold. Drug resistance monitoring and prevention are critical components of any national AIDS planning effort and can help guarantee the optimization of the entire diagnostic and treatment chain, providing patients with the best antiretroviral treatment regimens, raising the standard of HIV care, and ultimately ensuring the durability and long-term efficacy of ATR. Regular monitoring of programmed quality indicators related to treatment failure and/or drug resistance can reduce the possibility of the development and spread of HIV drug resistance.

## Figures and Tables

**Figure 1 viruses-16-00239-f001:**
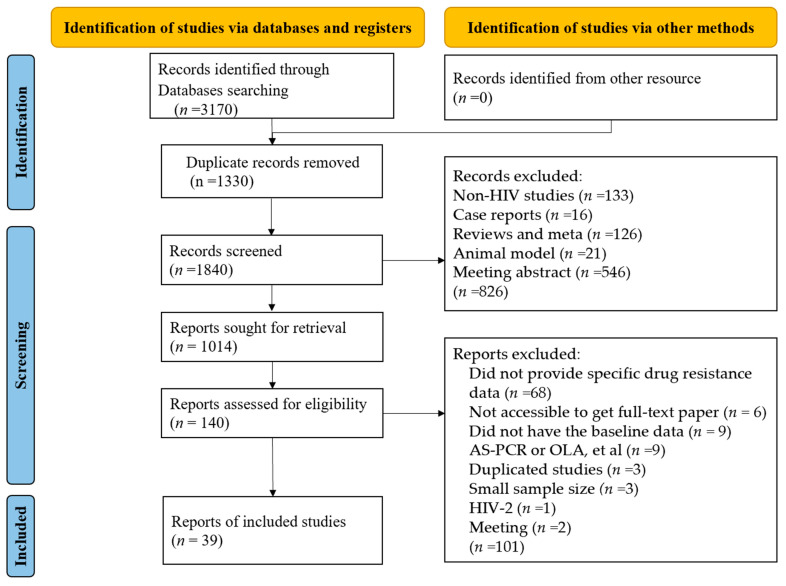
PRISMA 2020 study selection flow diagram for meta-analysis of HIV drug resistance detected by next-generation sequencing among ART-naïve patients.

**Figure 2 viruses-16-00239-f002:**
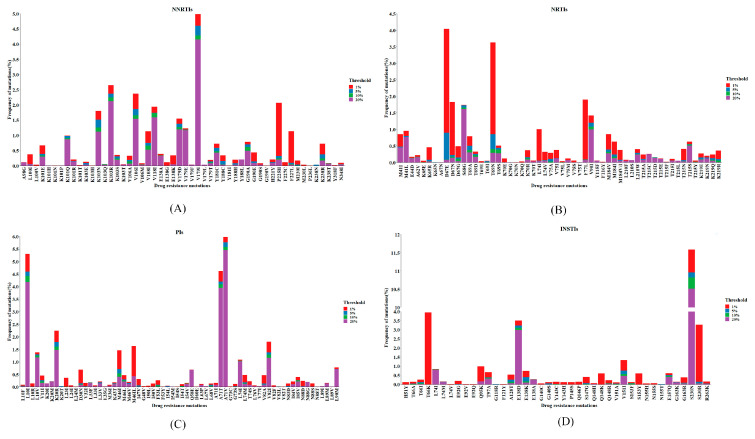
The pool prevalence of drug resistance mutations in patients with any mutation. (**A**) NNRTI-associated drug resistance mutations. (**B**) NRTI-associated drug resistance mutations. (**C**) PI-associated drug resistance mutations. (**D**) INSTI-associated drug resistance mutations.

**Figure 3 viruses-16-00239-f003:**
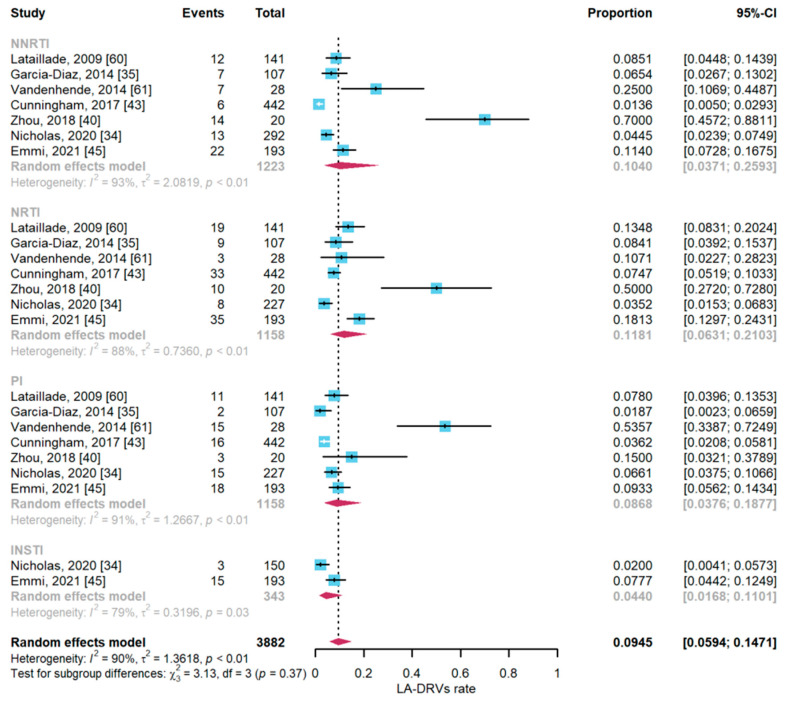
The prevalence of low-abundance pretreatment drug resistance.

**Table 1 viruses-16-00239-t001:** Summary of 39 studies on HIV drug resistance detected by NGS in ARV drug-naïve individuals.

Parameter	Studies, n (%)
**Number of participants, median (min–max)**	15,242, 148 (20–2902)
**Stage of HIV-1 infection at time of inclusion**	
Acute/recent	4 (10.3)
(Mainly) chronic	4 (10.3)
Not specified	31 (79.4)
**Type of specimen used for DRMs detection**	
Plasma	37 (94.9)
PBMC	1 (2.6)
Dried blood spot	1 (2.6)
**Geographic area**	
Europe	7 (17.9)
North America	6 (15.4)
Africa	13 (33.3)
Asia	6 (15.4)
Latin America	3 (7.7)
Worldwide	2 (5.1)
Not specified	2 (5.1)
**Study Purpose**	
TDR	14 (35.9)
PDR	17 (43.4)
LA-DRVs	8 (20.5)
**Detection method**	
454 pyrosequencing	14 (35.9)
Illumina NGS	23 (59.0)
tagged pooled pyrosequencing	1 (2.6)
Sentosa NGS system	1 (2.6)
**Data report**	
Only NGS data	27 (69.2)
SGA and NGS data	12 (30.8)
**Sample size**	
≤200	23 (59.0)
>200	16 (41.0)

**Table 2 viruses-16-00239-t002:** Pooled prevalence of pretreatment drug resistance at different thresholds.

Variants	Study	Event	Total	Pooled Prevalence (%) (95% CI)	I^2^ (%)	Q_B_	*p* Value
**PDR**				16.69 (13.53–20.41)	94.5	26.34	<0.01
1%	16	664	3202	29.74 (20.77–40.59)	96.6		<0.01
2%	6	581	2778	22.43 (18.81–26.53)	67.8		<0.01
5%	7	361	2524	15.47 (10.89–21.49)	90.4		<0.01
10%	3	103	830	12.95 (7.54–21.36)	91.7		<0.01
20%	20	852	6889	11.08 (8.43–14.43)	85.6		<0.01
**NNRTI**				9.54 (7.56–11.98)	88.7	14.59	<0.01
1%	14	289	2005	15.36 (10.84–21.32)	83.9		<0.01
2%	6	279	3229	9.51 (4.40–19.37)	91.9		<0.01
5%	7	354	4321	8.37 (6.90–10.11)	73.6		<0.01
10%	2	29	447	6.49 (4.55–9.18)	0.0		0.73
20%	13	581	7687	6.64 (4.39–9.91)	88.5		<0.01
**NRTI**				7.94 (2.76–5.80)	93.7	28.25	<0.01
1%	14	317	2088	14.94 (10.23–21.30)	89.9		<0.01
2%	6	301	3229	10.01 (7.35–13.50)	74.4		<0.01
5%	7	251	4321	6.06 (3.73–9.70)	90.3		<0.01
10%	2	29	447	6.49 (4.55–9.18)	32.0		0.23
20%	13	285	7687	4.01 (2.76–5.80)	81..8		<0.01
**PI**				4.78 (3.29–6.88)	93.8	82.9	<0.01
1%	13	182	1486	12.74 (8.14–19.40)	86.4		<0.01
2%	6	419	3951	9.49 (7.67–11.67)	47.3		0.09
5%	7	226	5042	2.92 (1.28–6.50)	93.3		<0.01
10%	2	26	447	5.72 (2.80–11.30)	84.1		0.01
20%	13	160	8468	1.72 (1.20–2.41)	38.4		0.08
**INSTI**				1.15 (0.51–2.55)	86.0	11.25	0.02
1%	4	29	764	3.71 (1.91–7.07)	72.3		<0.01
2%	2	50	1763	1.77 (0.56–5.48)	86.6		<0.01
5%	3	28	1811	1.93 (0.47–7.61)	89.5		<0.01
10%	1	1	425	0.24 (0.03–1.65)	– –	– –	– –
20%	4	24	4365	0.25 (0.03–2.35)	77.0		<0.01

**Table 3 viruses-16-00239-t003:** Subgroup analysis of pretreatment drug resistance.

Subgroup	Study	No. of Included	Event	Prevalence	95% CI (%)	I^2^ (%)	Q_B_	*p* Value
**1%**							0.43	0.5105
Illumina NGS	8	1384	385	25.79	16.96–37.16	92.8		
454	7	739	265	33.14	16.59–55.26	95.5		
**2%**							– –	– –
Illumina NGS	4	1013	212	21.66	17.33–26.72	75.3	12.16	
454	0	– –	– –	– –	– –	– –		
**5%**							0.27	0.6011
Illumina NGS	5	2293	296	10.42	4.40–22.69	62.3		
454	1	48	5	13.02	10.89–15.51	– –		
**10%**							12.11	0.0005
Illumina NGS	2	647	62	9.69	6.48–14.23	81.7		
454	1	183	41	22.40	16.94–29.01	– –		
**20%**							0.27	0.6012
Illumina NGS	13	6060	747	10.77	8.43–13.66	85.2		
454	5	409	61	12.72	7.05–21.89	88.0		

Event: The participants with any pretreatment drug resistance; 454: 454 pyrosequencing.

**Table 4 viruses-16-00239-t004:** The comparison between SGS and NGS.

Threshold	Variables	Subgroup	Study	Event	Total	95% CI (%)	I^2^ (%)	Q_B_	*p* Value
**20%**	PDR		19	419	3353	11.67 (8.62–15.79)	87.7	0.07	0.79
		SGS	12		2160	11.42 (7.46–17.48)	90.8		
		NGS	7		1193	12.36 (8.34–18.30)	77.4		
	NNRTI		18	212	3160	5.62 (3.86–8.18)	81.1	0.19	0.66
		SGS	11		1967	5.21 (3.24–8.38)	79.5		
		NGS	7		1193	6.22 (3.27–11.8)	81.5		
	NRTI		18	144	3160	4.57 (3.28–6.38)	0.693	0.46	0.5
		SGS	11		1967	4.20 (2.64–6.70)	72.9		
		NGS	17		1193	5.28 (3.33–8.38)	63.8		
	PI		16	208	2946	6.17 (4.22–9.01)	81.3	0.4	0.53
		SGS	10		2160	5.57 (3.43–9.05)	80		
		NGS	6		614	7.21 (3.84–13.56)	80.8		
**20%(SGS) vs. 1%**	PDR		16					13.93	<0.01
		SGS	12		2160	11.42 (7.46–17.48)	90.8		
		NGS	4		586	26.81 (23.32–30.82)	17.2		
	NNRTI		16		2581			6.46	0.01
		SGS	11		1967	5.21 (3.24–8.38)	79.5		
		NGS	5		614	11.87 (7.79–18.09)	64.6		
	NRTI		16					9.93	<0.01
		SGS	11		1967	4.20 (2.64–6.70)	72.9		
		NGS	5		614	12.33 (7.63–19.94)	76		
	PI		15		2474			2.72	0.1
		SGS	10		1860	3.24 (1.38–7.63)	95.1		
		NGS	5		614	9.74 (3.62–26.23)	95.1		

SGS: Sanger Sequencing, NGS: Next-generation Sequencing, PDR: Pretreatment drug resistant, PI: Protease Inhibitors, NNRTI: nonnucleoside reverse transcriptase inhibitors, NRTI: nucleoside reverse transcriptase inhibitors.

## Data Availability

All data used in this study were extracted from published papers and the detailed information is attached in the Appendix A.

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
