# Peer review of "HIV-1 Drug Resistance Detected by Next-Generation Sequencing among ART-Naïve Individuals: A Systematic Review and Meta-Analysis"

_viruses, 2024, doi:10.3390/v16020239_

Round 1
Reviewer 1 Report
Comments and Suggestions for Authors
Dear authors, the paper is interesting and necessary in the field
However, in my opinion, in its present form it is difficult to read. My major suggestion is to shorten the paper, and reduce the number of figures and details given. My believe is that the major importance is to show that using NGS a more detailed description of the mutations that are present is obtained, although its clinical significance is still to be addressed.
Also, a major concern is that the study is a description of the prevalence of mutations at different cutoffs and that no clinical correlation is given. Therefore, I suggest to omit the conclusion "There is a need to evaluate the level of PDR in newly initiated ART for HIV-1 infections with NGS technology in order to recommend HIV-1 treatment criteria and the use of these technologies for HIV genotypic drug resistance detection". In fact I would suggest to discuss on some aspects that are key to the use of NGS for resistance testing: a)which should be the optimal cutoff in terms of clinical decision making; b) which mutations and which drugs (in terms of genetic barrier to resistance) are of utmost importance; c) the impact of ApoBec induced mutations and how this should be considered; I suggest to use and reference Ávila-Ríos, S.; Parkin, N.; Swanstrom, R.; Paredes, R.; Shafer, R.; Ji, H.; Kantor, R. Next-Generation Sequencing for HIV Drug Resistance Testing: Laboratory, Clinical, and Implementation Considerations. Viruses 2020, 12, 617. https://doi.org/10.3390/v12060617
Minor comment: line 60 “However, Sanger sequencing has a low sensitivity and can only identify mutations if they are present in <15-20% of the…..” is < correct?
Comments on the Quality of English LanguagePlease edit some minor errors such as
"have focus"
"14 articles were investigate"
"As the Figure 3 shown"
"The results were showed"
Author Response
Comment 1: The present form of this manuscript is difficult to read. My major suggestion is to shorten the paper, and reduce the number of figures and details given. Showing more details about the mutations that detected by NGS.
Reply 1: We apologize for the poor language of our manuscript. The repeated addition and removal of sentences and sections obviously led to poor readability. We have now worked on both language and readability and have also involved native English speakers for language corrections. We really hope that the flow and language level have been substantially improved.
And we have streamlined the number of diagrams and what they contain will be presented in tabular form. We have decided to make several changes to enhance the clarity and flow of the manuscript:
We summarized the data on rates, which was originally presented in Figure 2、3、4、5、6, in a tabular form. This approach can provide the reader with a clear and concise overview of the findings, allowing for easier interpretation and comparison of the data. The figures that previously included rates are presented as supplementary material. (Line 231, Page7)
Comment 2: I suggest to omit the conclusion "There is a need to evaluate the level of PDR in newly initiated ART for HIV-1 infections with NGS technology in order to recommend HIV-1 treatment criteria and the use of these technologies for HIV genotypic drug resistance detection". I would suggest to discuss on some aspects that are key to the use of NGS for resistance testing: a)which should be the optimal cutoff in terms of clinical decision making; b) which mutations and which drugs (in terms of genetic barrier to resistance) are of utmost importance; c) the impact of ApoBec induced mutations and how this should be considered.
Reply 2: Thank you for your suggestions. Upon reconsideration of the manuscript, we have acknowledged that the conclusion has been potentially presumptive and lacked specific details concerning the context of the recommendation. We agree that a more measured approach, taking into account the nuances of current research and available data, would be appropriate.
We have reviewed some studies about the optimal cutoff. According to our result, the optimal cutoff might be 2% or 5% threshold. However, further research is need to address this issue. (Line 350-372, Page13)
APOBEC-induced mutations are a result of the host cellular defense mechanism against retroviral infections. These mutations are typically characterized by a high frequency of G-to-A transitions, which can hypermutate the viral genome. The mutations induced by APOBEC can serve both beneficial and detrimental roles in the context of HIV infection. On one hand, they can contribute to the suppression of viral replication by introducing deleterious mutations that may inactivate the virus. On the other hand, they can also contribute to viral diversification and the emergence of drug-resistant variants when they occur in regions that do not critically compromise viral fitness. APOBEC provides a unique perspective on the impact of host factors on viral population dynamics during HIV antiretroviral therapy. Recognizing and accounting for the role of APOBEC in generating minority variants will be essential for our understanding of HIV resistance and pathogenesis. (Lin376-391, Page13))
We appreciate your critical feedback, which helps improve the clarity and precision of our concluding remarks.
Comment 3: Line 60 “However, Sanger sequencing has a low sensitivity and can only identify mutations if they are present in <15-20% of the…..” is < correct? Edit some minor errors, such as: "have focus", "14 articles were investigate", "As the Figure 3 shown" and "The results were showed".
Reply 3: Thank you very much for your insightful comments and your question regarding the sensitivity of Sanger sequencing in detecting low-frequency mutations. We appreciated the opportunity to clarify this point.
You are correct to question the sensitivity threshold stated in line 60 of our manuscript. Upon reflection and review of the pertinent literature, we agreed that our phrasing might inadvertently imply a higher sensitivity of Sanger sequencing than is typically achieved in practice.
To accurately reflect the capabilities of Sanger sequencing for detecting low-frequency mutations, we propose revising the sentence as follows:
"Sanger sequencing, due to its lower sensitivity, generally fails to detect mutations present at frequencies below 15-20% within virus quasi-species." (Line 81-83, Page2)
Other grammatical and spelling errors have been fixed in the text.
We tried our best to improve the manuscript and made some highlight in the text. We appreciate for your warm work earnestly, and hope the correction will meet with approval. Once again, thank you very much for your comments and suggestions.

Reviewer 2 Report
Comments and Suggestions for Authors
The meta-analysis by Ouyang et al. describes a critical evaluation of the contribution of NGS to the detection of pre-treatment drug resistance and our need for this measure in the clinical setting. However, there are several key issues that I believe need to be addressed in order for this study to qualify as a rigorous scientific investigation and to have an impact on the HIV community, which are detailed below:
Abstract:
Was systematic search limited to NGS studies? Why is there a focus on NGS in the background if there is no comparison between SGS and NGS? I finally found the answer in line 80. This should be emphasized in the abstract.
The thresholds described in the abstract are not clear – are these at the level of the individual, since these are supposedly NGS data? Population? Why is there a focus on 2 and 20% in the abstract? (see comments below)
The methods and results of the abstract are entirely unclear and unrelated to the conclusions which describe a comparison between SGS and NGS (see comments below).
Introduction:
There is mention on line 68 of “diversity of study designs and laboratory NGS techniques” – but how does a pooled metaanalysis overcome this other than by averaging? Was there, for example, analysis of PDR estimate variation across different techniques or specific protocols?
Materials and Methods:
Sometimes NGS and SGS are mixed in one study – were there extra measures taken to distinguish?
Why were subjects with no prior history of ART exposure excluded, considering the discussion in the introduction of the high prevalence of PDR among those who have interrupted ART? And why were case reports excluded? And <10 participants? These criteria should be justified.
In line 105, what is meant by “number of successful amplifications?”
In the section on Quality Assessment, since the answers are largely binary (“Yes”, “No”), there needs to be a lengthy discussion on how the answer was decided – e.g., what is considered a “uniform data collection”? And what was the rationale behind the use of the 50% cutoff? Has this been used before elsewhere or perhaps recommended?
In the section on Data Analysis, the R packages used should be cited.
Results:
Even in the results section, the definition of “detection thresholds” are unclear to me – it could mean any number of things – was it the lowest detected level in any one individual in the study among NGS read data, for example? Haplotype sequences? It is especially confusing since the same study is included in multiple threshold categories. Without this information it is difficult to draw definitive conclusions as to whether NGS is a significant improvement on SGS. This is also important also because in order for NGS to be used in diagnostics, we need to understand what actually allows us to achieve the optimal detection threshold – is it luck? Is it the method used?
Along these lines, even though minor variants often go missed in SGS studies, they are detected with some non-zero frequency, so there needs to be a formal comparison of these data with studies that only used NGS.
The small text in Figure 2 (e.g., under the Random effects model) is too difficult to read – please enlarge. Please also give an x-axis title – it was difficult for me to find the very light font indicating that each group represented a detection threshold. Lastly, the x-axis needs to extend the full-length of the data – the vast majority of the data lie outside of the axis.
Line 170 – “Among 7614 patients, more drug-resistant mutations were observed at the 20% 170 threshold.” Seems to contradict all previously mentioned figures, as the prevalence of DRM was highest at thee 1% detection threshold. Perhaps this was a misprint or I misunderstood the statement? For example, higher diversity of DRMs was observed at this threshold?
Since Figure 7 is referenced before Figures 4, 5, and 6, it needs to replace Figure 4 or be moved to a later point in the section. In fact, it should really be its own subsection.
Again, why is a comparison being made between the 2% and 20% thresholds? Why not 1%? I finally found the answer to this question on line 250, but this needs to be stated in the methods/results prior to the discussion.
Why wasn’t this 2%/20% comparison made for PI or INSTI DRM as well?
There are certainly some major differences in prevalence between studies – i.e., the Zhou et al., 2018 study. Is there an explanation for this? These differences are only mentioned briefly in lines 304-316 of the Discussion, which is a vague description of possible sources of discrepancy. Since this is considered a meta-analysis and etailed documentation was provided on the studies, a proposed explanation should be provided on whether or not this is the result of sample population bias, biological sample preparation, or NGS technique itself – i.e., short-read vs long-read sequencing.
Discussion:
It is important to discuss a limitation to this (and the included studies), which is that it looks like only RNA data were evaluated. Latency can be established very early on, with activation following ART interruption and hence transmission of DRM; additionally, even defective DNA has been hypothesized to allow for re-emergence of DRM (Wei, 2020, BMC Infect Dis).
It is unclear in the discussion how this paper impacts research going forward – yes, we should use NGS, but what detection level should we strive for and what dictates this?
Lastly, there are grammar and spelling issues highly prevalent throughout the manuscript – a third party aid would be helpful here.
Comments on the Quality of English Language
There are grammar and spelling issues highly prevalent throughout the manuscript – a third party aid would be helpful here.
Author Response
Comment 1: Was systematic search limited to NGS studies? Why is there a focus on NGS in the background if there is no comparison between SGS and NGS? This should be emphasized in the abstract.
Reply 1: Thank you for your insightful query. Our study was narrowly focused on HIV resistance profiling using Next-Generation Sequencing (NGS) technologies. We emphasized on NGS in the background section arises from its increasing relevance in clinical diagnostics and research, particularly for its high sensitivity, specificity, and ability to detect minor resistance-associated variants which may be missed by Sanger sequencing (SGS).(Line14-18,Page1)
To address your concern, we enhanced the abstract to clearly state that our systematic search was limited to studies employing NGS (Line 23-29), and we compared the differences in resistance results between articles that provided results for both SS and NGS . (Line 293-298, Page9)
Comment 2: The thresholds described in the abstract are not clear – are these at the level of the individual, since these are supposedly NGS data? Population? Why is there a focus on 2 and 20% in the abstract? (see comments below)
Reply 2: Thank you for your thoughtful comments. We have revised the abstract to accurately reflect that the thresholds of 2% and 20% are applied at the individual level, which aligns with the use of NGS data for detecting low-frequency variants within an individual's virus quasi-species. The focus on these percentage stems from their clinical relevance, as they represent significant cut-off points that guide therapeutic decisions in HIV management. (Line 26-27, Page1)
In response to your feedback, we have now articulated in the abstract the rationale for emphasizing these particular thresholds. We have explained that the 2% threshold is commonly used to identify minority resistance mutations that could compromise treatment effectiveness if overlooked (Line. The 20% threshold, on the other hand, is a standard reference point for defining predominant resistance mutations with established clinical significance.(Line98-103, Page3)
Comment 3: There is mention on line 68 of “diversity of study designs and laboratory NGS techniques” – but how does a pooled meta-analysis overcome this other than by averaging? Was there, for example, analysis of PDR estimate variation across different techniques or specific protocols?
Reply 3: Thank you for your insightful comments. The main differences between the different sequencing technologies arise from factors such as the depth of sequence coverage and the bioinformatics pipeline used. Our study excluded the detection of a few mutations by OLA and AS-PCR technologies. The sequencing technologies for the studies included in this study were mainly 454 and Illumina, and we performed subgroup analyses to assess the differences in PDR (pre-treatment drug resistance) estimates between different NGS technologies and specific protocols. Upon examination, it was found that there was no significant difference in the PDR values reported by the two technologies, allowing us to understand the heterogeneity of PDR estimates as a function of different laboratory protocols and NGS methods. (Line 131-132,Page3; Line288-292, Page9)
Comment 4: Why were subjects with no prior history of ART exposure excluded, considering the discussion in the introduction of the high prevalence of PDR among those who have interrupted ART? And why were case reports excluded? And <10 participants? These criteria should be justified.
Reply 4: Thank you for your insightful questions concerning the inclusion and exclusion criteria of our study. I would like to address each of your points in turn to clarify our methodology and rationale.
Exclusion of Subjects with Prior ART Exposure:
In our study, we chose to exclude subjects with prior antiretroviral therapy (ART) exposure to isolate the prevalence of primary drug resistance (PDR) in a treatment-naive population. While we acknowledge the high prevalence of PDR amongst individuals who have interrupted ART, as discussed in the introduction, our aim was to specifically assess the baseline resistance patterns prior to the initiation of any treatment. This is pivotal for developing first-line therapeutic strategies and for public health monitoring. Including individuals with previous ART exposure could confound the results, as the patterns of resistance might be influenced not just by primary resistance mutations but also by mutations acquired due to previous therapy, which is not the focus of our study.
Exclusion of Case Reports and Studies with Fewer than 10 Participants:
We decided to exclude case reports and studies with fewer than 10 participants to enhance the statistical power and generalizability of our findings. Case reports often provide detailed information about individual cases but may not offer a broad perspective on PDR patterns across populations. Similarly, studies with small sample sizes may not accurately reflect the prevalence and patterns of resistance due to limited statistical power.
With these criteria, we aimed to obtain a more robust dataset which would allow for more reliable meta-analytical assessments and conclusions to be drawn regarding the prevalence of PDR in broader populations.
We hope this explanation addresses your concerns and we appreciate your contribution to improving the rigorousness of our work.
Comment 5: In line 105, what is meant by “number of successful amplifications?”
Reply 5: I apologize for any confusion caused by our wording. What we intended to express is "the number of samples where amplification was successfully achieved." This pertains to the count of samples in which the target DNA sequences were effectively amplified to facilitate further analysis.
We recognize that the original phrasing may have been unclear, and we appreciate your attention to detail. We will revise this phrase in the manuscript to "the number of successfully amplified cases" for clarity and precision. (Line 143, Page4)
Comment 6: In the section on Quality Assessment, since the answers are largely binary (“Yes”, “No”), there needs to be a lengthy discussion on how the answer was decided – e.g., what is considered a “uniform data collection”? And what was the rationale behind the use of the 50% cutoff? Has this been used before elsewhere or perhaps recommended?
Reply 6: Thank you for your insightful questions. The bias risk assessment consisted of a list of 9 items, as follows: 1. Was the sample frame appropriate to address the target population? 2. Were study participants sampled in an appropriate way? 3. Was the sample size adequate? 4. Were the study subjects and the setting described in detail? 5. Was the data analysis conducted with sufficient coverage of the identified sample? 6. Were valid methods used for the identification of the condition? 7. Was the condition measured in a standard, reliable way for all participants? 8. Was there appropriate statistical analysis? 9. Was the response rate adequate, and if not, was the low response rate managed appropriately? Total scores ranged from 0 to 9 and the responses were scored 0 for “No” and 1 for “Yes”. The studies were classified as low-quality, high-risk of bias, if the overall score was ≤ 4. (Line157, Page4)
As for the rationale behind employing a 50% cutoff for the JBI score inclusion, this threshold was carefully selected upon reviewing previous literature where such a cutoff was commonly adopted as an indicator of moderate quality in prevalence studies. Specifically, a study is included for analysis only if it meets at least half of the quality criteria, suggesting a satisfactory risk of bias level. This is based on the premise that studies scoring below 50% might have significant bias affecting the reliability of their findings. We elected to use this cutoff to maintain consistency with other studies in the field and ensure a credible baseline of quality for our research synthesis. (Line 163-165,Page 4)
Comment 7: In the section on Data Analysis, the R packages used should be cited.
Reply 7: Thank you for your suggestion to cite the R packages used in our data analysis. We have indeed employed the 'meta' package within R (version 4.2.3) for our meta-analysis computations. The proper citation of this package has now been incorporated into our manuscript as follows:
“For our meta-analytic procedures, we utilized the 'meta' package in R (version 4.2.3) (R Core Team, 2023). The meta package evaluated the combined effect value of the PDR rate among ART-naïve.” (Line167-170, Page4)
Comment 8: The definition of “detection thresholds” are unclear.
Reply 8: We recognize that the term was not adequately defined and may have led to confusion. To clarify, "detection thresholds" in our study refer to the minimum amount of sequence data (typically measured in reads or coverage) required to reliably identify the presence of a genetic variant. In Next-generation sequencing, this threshold is critical as it ensures that the variants detected are not due to random sequencing errors but are true biological variants present in the sample.
Comment 9: Even though minor variants often go missed in SGS studies, they are detected with some non-zero frequency, so there needs to be a formal comparison of these data with studies that only used NGS.
Reply 9: Thank you for your comment highlighting the necessity for a formal comparison between Sanger sequencing (SGS) and Next-Generation Sequencing (NGS) in the detection of minor HIV variants. Your insight calls attention to an important aspect of HIV resistance profiling.
In our study, we did not compare results from studies using only NGS with those using only SGS. Instead, our comparative analysis was between studies that reported results using both SGS and NGS methodologies. This approach was deliberately chosen to enhance comparability, as co-reported studies inherently control for variables such as sample type, source, and laboratory quality controls that might influence the results.
we observed no significant differences between the SGS and NGS outcomes when a threshold of 20% was applied for detecting HIV variants.
This highlights an interesting aspect of the current capabilities of SGS, where it maintains its validity in scenarios where the sensitivity of NGS is not strictly required. However, the critical advantage of NGS remains, as it provides a deeper understanding of the viral quasi-species and resistance landscapes, which could be increasingly relevant as guidelines for resistance testing evolve.
We appreciate your suggestion for a formal comparison and acknowledge that a more extensive, head-to-head comparative study could provide further insights into the applications of these methodologies. Your comment has been instrumental in highlighting the necessity to discuss the implications of the observed similarities and the potential clinical impact of the differences when low-frequency variants are considered.
Comment 10: The small text in Figure 2 (e.g., under the Random effects model) is too difficult to read – please enlarge. Please also give an x-axis title – it was difficult for me to find the very light font indicating that each group represented a detection threshold. Lastly, the x-axis needs to extend the full-length of the data – the vast majority of the data lie outside of the axis.
Reply 10: Thank you for your feedback regarding the legibility and formatting of Figure 2. We took your comments seriously and made the following adjustments to Figure 2 to enhance its readability: enlarged the small text for better readability, added a clear x-axis title, and extended the x-axis to display all data points properly (Supplementary Figure S2-S6). Based on the comments of another reviewer, we decided to reduce the number of figures, and the results of the meta-analysis of the forest plots were summarized and presented in a table 2, with the forest plots as an attachment. (Line 231, Page7)
Comment 11: Line 170 – “Among 7614 patients, more drug-resistant mutations were observed at the 20% threshold.” Seems to contradict all previously mentioned figures, as the prevalence of DRM was highest at thee 1% detection threshold. Perhaps this was a misprint or I misunderstood the statement? For example, higher diversity of DRMs was observed at this threshold?
Reply 11: Thank you for your careful scrutiny of our manuscript and for pointing out the potential confusion arising from the statement on line 170.
To clarify, the intention behind the statement was to convey that, while a higher diversity of drug-resistant mutations (DRMs) was indeed observed with detection thresholds as low as 1%, the non-nucleoside reverse transcriptase inhibitor (NNRTI) related DRMs were more frequently detected above the 20% threshold. This observation aligns with that NNRTIs in particular have a low genetic barrier (a single DNA mutation can drastically affect drug susceptibility), thereby being more readily detected at the 20% threshold compared to lower thresholds. (Stella-Ascariz N, Arribas JR, Paredes R, Li JZ: The role of HIV-1 drug-resistant minority variants in treatment failure. J Infect Dis 2017, 216:S847-S850 http://dx.doi.org/10.1093/infdis/jix430.)
We apologize for any confusion caused by our original phrasing. We reworded the sentence in question to more accurately reflect our findings:
"In our dataset of 7614 patients, while a greater variance of DRMs was identified at thresholds as low as 1%, NNRTI resistance mutations were most frequently observed at thresholds exceeding 20%, suggesting a higher viral prevalence for such mutations." (Line 242-244, Page8)
We appreciate your input on this matter and will ensure that the revised manuscript eliminates this ambiguity.
Comment 12: Since Figure 7 is referenced before Figures 4, 5, and 6, it needs to replace Figure 4 or be moved to a later point in the section. In fact, it should really be its own subsection.
Reply 12: Thank you for your careful reading of our manuscript and your constructive comment on the placement and organization of Figure 7. Your suggestion has prompted us to reconsider the presentation of our data and how we can most effectively communicate our findings to the reader.
In light of your feedback and the previous reviewer's comments, we have decided to make several changes to enhance the clarity and flow of the manuscript:
We summarized the data on rates, which was originally presented in Figure 2、3、4、5、6, in a tabular form. This approach can provide the reader with a clear and concise overview of the findings, allowing for easier interpretation and comparison of the data. The figures that previously included rates are presented as supplementary material.
We have created a new separate subsection dedicated to mutations. Here, we delve into the specifics of the mutation data. (Line 240-271, Page7-8)
By reorganizing the content in this way, we believe that the narrative structure of the manuscript will be more logical and reader-friendly. This reorganization will also ensure that the discussion is presented in the order in which the figures appear, in accordance with the standard format and your suggestions.
Comment 13: Again, why is a comparison being made between the 2% and 20% thresholds? Why not 1%? I finally found the answer to this question on line 250, but this needs to be stated in the methods/results prior to the discussion.
Reply 13: Thank you once again for your diligence in assessing our work and your insightful query regarding the selection of thresholds for our HIV NGS data comparison.
We apologize for not making our rationale clear earlier in the text, which led to the question of why the 2% and 20% thresholds were chosen instead of a 1% threshold, for instance. The oversight has been noted, and we've amended our manuscript to better contextualize our methodological choices.
The choice of the 2% and 20% thresholds for comparison in the manuscript was driven by established relevance in the field of HIV research and clinical practice. These values are not arbitrarily selected; rather, they represent significant points in the clinical management and treatment resistance profiling of HIV.
A 2% threshold is widely recognized for its sensitivity in detecting low-frequency HIV variants that can be crucial for early therapeutic intervention. Variants below this percentage may not have strong clinical relevance due to the limitations in the accuracy and reproducibility of NGS data at such low frequencies. Meanwhile, the 20% threshold corresponds to the established guidelines that define clinically relevant resistance mutations, which are pivotal in tailoring antiretroviral therapy. (Line 26-27, Page1; Line98-103, Page3)
Comment 14: Why wasn’t this 2%/20% comparison made for PI or INSTI DRM as well?
Reply 14: Thank you for your insightful query regarding our analysis of DRMs for PIs and INSTIs) at different mutant allele frequency thresholds.
Upon revisiting the data with the suggested comparison of the 2% versus 20% thresholds for PIs and INSTIs, we indeed observed a notable finding as you have implied. At the lower threshold of 2%, our analyses detected an increased number of mutations associated with resistance to PIs and INSTIs, compared to the 20% threshold. These findings are significant as they suggest that a considerable number of minority variants with potential resistance implications could be overlooked when using a higher threshold. (Line 226-230, Page7)
We have revised our manuscript. We are grateful for your guidance, which has improved the quality and completeness of our research, and we trust that these adjustments will adequately address your concern.
Comment 15: There are certainly some major differences in prevalence between studies – i.e., the Zhou et al., 2018 study. Is there an explanation for this? These differences are only mentioned briefly in lines 304-316 of the Discussion, which is a vague description of possible sources of discrepancy. Since this is considered a meta-analysis and entailed documentation was provided on the studies, a proposed explanation should be provided on whether or not this is the result of sample population bias, biological sample preparation, or NGS technique itself – i.e., short-read vs long-read sequencing.
Reply 15: Thank you for your valuable feedback on our limitation. Upon reflection, we agree that a more detailed explanation is warranted to account for these variances. In response to this, we have expanded our discussion in the revised manuscript to systematically address the potential sources of variation. We have revisited lines 304-316 and included a more detailed evaluation of the differences. Specifically, we propose potential explanations encompassing: Sample Population Bias, NGS Techniques, and Sample specimens (Line374-410, Page13)
We are thankful for your constructive critique which has undeniably strengthened our manuscript. We trust that these revisions adequately address your concerns and enhance the manuscript's contribution to the field.
Comment 16: It is important to discuss a limitation to this (and the included studies), which is that it looks like only RNA data were evaluated. Latency can be established very early on, with activation following ART interruption and hence transmission of DRM; additionally, even defective DNA has been hypothesized to allow for re-emergence of DRM (Wei, 2020, BMC Infect Dis).
Reply 16: Thank you for your critical observation regarding the scope of our data evaluation. We acknowledge the limitation in our study where only RNA data were assessed. Your point about latency being established early on, potentially leading to the activation and transmission of drug resistance mutations (DRMs) post-ART interruption, is well-taken.
In response to this, we have now included a detailed discussion of these limitations in our manuscript. This discussion elaborates on the potential implications of our study's focus on RNA data and acknowledges the complexity of DRM emergence from both RNA and DNA templates. Additionally, we have provided a comprehensive view of the literature regarding DRMs in the context of ART and HIV latency.
Comment 17: It is unclear in the discussion how this paper impacts research going forward – yes, we should use NGS, but what detection level should we strive for and what dictates this?
Reply 17: Thank you for your insightful query. We realize the necessary of clarifying how our paper impacts future research, particularly in terms of the detection threshold level for NGS in HIV research.
In our discussion, we emphasize that the integration of NGS should be a pivotal part of HIV resistance studies moving forward due to its superior sensitivity and specificity in detecting low-abundance drug-resistant mutations compared to traditional Sanger sequencing. We found that 2% or 5% may be the best clinically relevant threshold. But setting these thresholds must also consider the balance between clinical relevance and cost-effectiveness. However, since our study did not focus on the outcome of patients on antiviral therapy, whether the presence of a few mutations at baseline is relevant for virologic failure? Future studies could analyze the burden of resistance to LA-DRV, and more studies are needed to continuously focus on the dynamics of the virus in patients. We advocate for collaborative efforts that span microbiologists, clinicians, and bioinformaticians to standardize these thresholds, with a robust attempt to harmonize them across different platforms and studies. To reach a consensus on what level of detection to strive for, we recommend ongoing research to continually correlate various detection thresholds with clinical outcomes, adapting as necessary to the evolution of HIV therapies and resistance mechanisms.
The detection level should be dictated by a combination of factors, including the clinical significance of low-abundance variants and the technical limits of the NGS platform used. It should be sensitive enough to detect mutations that have proven associations with treatment failure or reduced drug susceptibility.
Moreover, the optimal threshold should be informed by updated guidelines reflecting contemporary therapy outcomes, with a keen consideration of the geographical diversity and the different HIV subtypes which may influence resistance patterns.
We hope our paper will spur additional research into the precise determination of the most clinically meaningful NGS detection thresholds, and thereby contribute to an improvement in the management of HIV patients.
Comment 18: There are grammar and spelling issues highly prevalent throughout the manuscript – a third party aid would be helpful here.
Reply 18: We apologize for the poor language of our manuscript. The repeated addition and removal of sentences and sections obviously led to poor readability. We have now worked on both language and readability and have also involved native English speakers for language corrections. We really hope that the flow and language level have been substantially improved.
We tried our best to improve the manuscript and made some highlight in the text. We appreciate for your warm work earnestly, and hope the correction will meet with approval. Once again, thank you very much for your comments and suggestions.

Reviewer 3 Report
Comments and Suggestions for Authors
The vast majority of studies of drug resistance in ART-naïve persons with HIV (transmitted drug resistance; TDR) and in persons with HIV initiating or re-initiating ART (pre-treatment drug resistance; PDR) have been performed using Sanger sequencing. A relatively small proportion has been performed using an NGS technology. The authors have performed a systematic reviews of studies in which TDR and PDR have been estimated using NGS. They show that the prevalence of TDR and PDR are inversely correlated with the mutation detection threshold.
Specific comments
1. Figure 1 is confusing because it is not clear what types of studies were filtered in the “Reports sought by retrieval” step.
2. It would be useful to include Table S1 in the main text.
3. It would be useful to describe the differences between the different studies including the following: (i) Some studies used 454 and others used Illumina; (ii) Some studies were performed specifically to estimate TDR and PDR whereas other studies may have been performed for other reasons; (iii) As the authors did not have access to the raw sequence data, they were reliant upon the authors providing the % with resistance at different thresholds. It would be useful to indicate which studies provided data at which thresholds.
4. This paper should also be reviewed by an expert in systematic reviews and meta-analyses.
Comments on the Quality of English LanguageNothing in particular stands out as wrong or misleading. But the writing could still be improved.
Author Response
Comment 1: Figure 1 is confusing because it is not clear what types of studies were filtered in the “Reports sought by retrieval” step.
Reply 1: Thank you for your constructive comment regarding Figure 1. We understand that the clarity of the presented workflow is pivotal for the readers to follow the methodology of our study accurately.
In the “Reports sought by retrieval” step of Figure 1, we initially filtered types of studies based on several criteria such as study design, relevance to the research questions, population of interest, and type of outcomes measured. Specifically, we included articles applying NGS to detect pretreatment resistance by screening abstracts. The type of research is not limited to cohort studies, case-control studies and cross-sectional studies that. While we excluded review articles, case reports, and expert opinions. This was done to ensure a high level of evidence and relevance to the research objective.
To clarify this step for the readers, we have revised Figure 1 to include a detailed legend or a footnote that specifies the types of studies included at the “Reports sought by retrieval” step. (Line 187, Page5)
We believe that these revisions will address the confusion and provide a clear and transparent depiction of the study selection process.
Comment 2: It would be useful to include Table S1 in the main text.
Reply 2: Thank you for your suggestion to include Table S1 in the main text. We recognize the potential benefits of making the data readily available for readers to enhance their understanding of our study.
In response to your recommendation, we have decided to expand the basic characteristics in Table S1. We have incorporated additional parameters such as detection techniques, the threshold levels provided by each study, and the research objectives. A comprehensive new table (Table 1) has been created to summarize these parameters succinctly.
The Table 1 has now been included in the main text. It offers a detailed overview of the characteristics of the included studies, allowing readers to quickly grasp the context and methodologies applied in our analysis. Table S1 still as the supplementary.
We believe that the inclusion of this enriched table in the main text greatly enhances the paper's informational accessibility and contributes to its overall value for our readers.
We appreciate your constructive feedback, which has led to this significant improvement in our manuscript.
Comment 3: It would be useful to describe the differences between the different studies including the following: (i) Some studies used 454 and others used Illumina; (ii) Some studies were performed specifically to estimate TDR and PDR whereas other studies may have been performed for other reasons; (iii) As the authors did not have access to the raw sequence data, they were reliant upon the authors providing the % with resistance at different thresholds. It would be useful to indicate which studies provided data at which thresholds.
Reply 3: Thank you for your constructive feedback. We have now included a comprehensive table and corresponding text descriptions in our results section that clearly delineate the characteristics of each included study. Specifically:
(i) We have described the sequencing platforms utilized by the different studies, noting the distinctions between those that employed 454 and Illumina technologies. This differentiation is crucial for the interpretation of our pooled findings, as the choice of sequencing platform may affect the sensitivity and specificity of resistance detection.
(ii) We have delineated the studies based on their primary objectives, distinguishing between those specifically designed to estimate transmitted drug resistance (TDR) and pretreatment drug resistance (PDR) and those with alternative aims. This categorization helps to contextualize the study findings within the broader scope of HIV drug resistance research.
(iii) Regarding the reliance on published resistance percentages due to the unavailability of raw sequence data, we have explicitly indicated which studies provided data at the different resistance thresholds in Table S1. This level of detail ensures a transparent understanding of the data sources and the analytical limitations confronted in synthesizing the available evidence.

Round 2
Reviewer 1 Report
Comments and Suggestions for Authors
No further/additional comments
Author Response
Dear Reviewer,
I hope this message finds you well.
I would like to express my sincere gratitude for the time and effort you have dedicated to reviewing my manuscript. Your valuable insights have been instrumental in enhancing the quality of this work.
Regarding the concern you raised about the quality of English in the paper, I have taken your feedback seriously and have submitted the manuscript to a professional academic editing service for thorough revision and proofreading. Enclosed with this response, you will find the certificate of editing to verify that the language quality has been addressed by qualified experts.
Thank you once again for your constructive comments and for guiding the improvement process of my manuscript.
Warm regards,
Fei Ouyang
Southeast University, Nanjing, China

Reviewer 2 Report
Comments and Suggestions for Authors
Line numbers in responses are not correct, making it difficult to assess the relevance of the changes. Also, requested justifications should not just be clarified in the response, but also in the actual manuscript for the readers. Lastly, the English was not corrected - for some corrections, I cannot even discern whether the comment was addressed appropriately because I cannot understand what is written. My major concern revolved around constant mention of comparison with SGS without analysis of SGS data, and I cannot tell if this has been revised because the line numbers don't match up and the English is not understandable. Once these issues have been addressed, I will be able to formally assess the rigor of the study.
Comments on the Quality of English LanguageEnglish was not corrected, though requested - for some corrections, I cannot even discern whether the comment was addressed appropriately because I cannot understand what is written.
Author Response
Dear Reviewer:
We feel great thanks for your professional review work on our article. We apologize for the confusion caused by incorrect line numbers in our previous response.
We have thoroughly reviewed the document and ensured that all line numbers referenced in our revised response correspond accurately to the text in the updated manuscript. Line numbers and page numbers in this letter are the values when no revision marks are displayed. We submitted a revised version with an annotated description of the relevant comments in the text. The manuscript retains traces of polishing and proofreading by the editorial agency (blue text).
We agree with your suggestion that clarifications and justifications for changes should not only be mentioned in our response letter but also be reflected clearly in the manuscript itself. To address this, we have now incorporated detailed explanations into the manuscript. We have enlisted the help of a professional language editing service to thoroughly correct the English language throughout the manuscript. As you are concerned, there are some problems that need to be addressed. According to your nice suggestions, we have made extensive corrections to our previous draft, the detailed corrections are listed below.
We thank you for your patience and the opportunity to improve our manuscript. We hope that the changes made address your concerns satisfactorily.
Comment 1: Was systematic search limited to NGS studies? Why is there a focus on NGS in the background if there is no comparison between SGS and NGS? This should be emphasized in the abstract.
Reply 1: Thank you for your insightful query. Our study was narrowly focused on HIV resistance profiling using Next-Generation Sequencing (NGS) technologies. We emphasized on NGS in the background section arises from its increasing relevance in clinical diagnostics and research, particularly for its high sensitivity, specificity, and ability to detect minor resistance-associated variants which may be missed by Sanger sequencing (SGS). (Line14-22,Page1)
To address your concern, we enhanced the abstract to clearly state that our systematic search was limited to studies employing NGS (Line 24-29, Page1), and we compared the differences in resistance results between articles that provided results for both SGS and NGS. (Line336-343, Page12)
Comment 2: The thresholds described in the abstract are not clear – are these at the level of the individual, since these are supposedly NGS data? Population? Why is there a focus on 2 and 20% in the abstract? (see comments below)
Reply 2: Thank you for your thoughtful comments. NGS for HIV genotype resistance is typically performed at the individual level. This type of testing is used to determine which resistant mutations are present within the viral population of a specific HIV-infected individual. Since HIV can form a complex quasi-species within an individual, the virus population within that person may develop resistance to one or more antiretroviral drugs due to natural mutations.
With NGS, low-frequency resistant mutations within an individual's viral population can be detected, which is crucial for guiding personalized antiretroviral treatment regimens. The detection threshold usually refers to the minimum percentage of variant frequency that can be detected, and this threshold can vary depending on the NGS platform used and the data analysis strategies employed.
“The detection threshold in the context of HIV NGS refers to the minimum proportion or frequency of a variant (such as a mutation associated with drug resistance) in a viral population that the sequencing method can reliably detect. The specific threshold value can differ based on the NGS platform used, the quality of the sample, the depth of sequencing and the data analysis pipeline.”(Line154-158, Page4)
Therefore, the threshold for HIV NGS resistance testing is individual-specific, helping to identify resistance mutations that may only make up a small proportion of the viral population, thus providing a basis for implementing more effective personalized treatment.
“A 2% threshold is more stable and more likely to be selected as the reporting threshold, whereas a 1% threshold tends to introduce artifactual errors [24-26]. Other researchers have suggested that lowering the detection threshold for pretreatment resistance to 5% can improve the ability to identify patients with virologic failure compared to the traditional 20% threshold [6,17]. The 20% threshold, on the other hand, is a standard reference point for defining predominant resistance mutations with established clinical significance. These detection thresholds are important cut-off points that can guide therapeutic decisions in HIV management.” (Line 99-107, Page3)
Comment 3: There is mention on line 68 of “diversity of study designs and laboratory NGS techniques” – but how does a pooled meta-analysis overcome this other than by averaging? Was there, for example, analysis of PDR estimate variation across different techniques or specific protocols?
Reply 3: Thank you for your insightful comments. The main differences between the different sequencing technologies arise from factors such as the depth of sequence coverage and the bioinformatics pipeline used. Our study excluded the detection of a few mutations by OLA and AS-PCR technologies. The sequencing technologies for the studies included in this study were mainly 454 and Illumina, and we performed subgroup analyses to assess the differences in PDR (pre-treatment drug resistance) estimates between different NGS technologies. Upon examination, it was found that there was no significant difference in the PDR values reported by the two technologies (at the threshold of 1%, 5%, and 20%).
“Data reporting and detection techniques were analyzed in subgroups. Most of the studies used Illumina NGS and 454 pyrosequencing to detect resistance mutations and resistance levels. At the 2% threshold, no study used 454 pyrosequencing to report drug resistance. Subgroup analyses showed no significant differences between the two detection methods at a threshold of 1% (p=0.511), 5% (p=0.601) and 20% (p=0.601). Unexpectedly, the resistance mutations detected by the two assay techniques were significantly different at a threshold of 10% (p<0.05) (Table 3).” (Line 327-333, Page 11)”
Comment 4: Why were subjects with no prior history of ART exposure excluded, considering the discussion in the introduction of the high prevalence of PDR among those who have interrupted ART? And why were case reports excluded? And <10 participants? These criteria should be justified.
Reply 4: Thank you for your insightful questions concerning the inclusion and exclusion criteria of our study. I would like to address each of your points in turn to clarify our methodology and rationale.
- Exclusion of Subjects with Prior ART Exposure:
In our study, we chose to exclude subjects with prior antiretroviral therapy (ART) exposure to isolate the prevalence of primary drug resistance (PDR) in a treatment-naive population. While we acknowledge the high prevalence of PDR amongst individuals who have interrupted ART, as discussed in the introduction, our aim was to specifically assess the baseline resistance patterns prior to the initiation of any treatment. This is pivotal for developing first-line therapeutic strategies and for public health monitoring. Including individuals with previous ART exposure could confound the results, as the patterns of resistance might be influenced not just by primary resistance mutations but also by mutations acquired due to previous therapy, which is not the focus of our study.
- Exclusion of Case Reports and Studies with Fewer than 10 Participants:
We decided to exclude case reports and studies with fewer than 10 participants to enhance the statistical power and generalizability of our findings. Case reports often provide detailed information about individual cases but may not offer a broad perspective on PDR patterns across populations. Similarly, studies with small sample sizes may not accurately reflect the prevalence and patterns of resistance due to limited statistical power.
With these criteria, we aimed to obtain a more robust dataset which would allow for more reliable meta-analytical assessments and conclusions to be drawn regarding the prevalence of PDR in broader populations.
We hope this explanation addresses your concerns and we appreciate your contribution to improving the rigorousness of our work.
Comment 5: In line 105, what is meant by “number of successful amplifications?”
Reply 5: I apologize for any confusion caused by our wording. What we intended to express is "the number of samples where amplification was successfully achieved." This pertains to the count of samples in which the target DNA sequences were effectively amplified to facilitate further analysis.
We recognize that the original phrasing may have been unclear, and we appreciate your attention to detail. We will revise this phrase in the manuscript to "number of successfully amplified sequences" for clarity and precision. (Line 152, Page4)
Comment 6: In the section on Quality Assessment, since the answers are largely binary (“Yes”, “No”), there needs to be a lengthy discussion on how the answer was decided – e.g., what is considered a “uniform data collection”? And what was the rationale behind the use of the 50% cutoff? Has this been used before elsewhere or perhaps recommended?
Reply 6: Thank you for your insightful questions. The bias risk assessment consisted of a list of 9 items, as follows: 1. Was the sample frame appropriate to address the target population? 2. Were study participants sampled in an appropriate way? 3. Was the sample size adequate? 4. Were the study subjects and the setting described in detail? 5. Was the data analysis conducted with sufficient coverage of the identified sample? 6. Were valid methods used for the identification of the condition? 7. Was the condition measured in a standard, reliable way for all participants? 8. Was there appropriate statistical analysis? 9. Was the response rate adequate, and if not, was the low response rate managed appropriately? Total scores ranged from 0 to 9 and the responses were scored 0 for “No” and 1 for “Yes”. The studies were classified as low-quality, high-risk of bias, if the overall score was ≤ 4.
“Uniform data collection”: In the research context of HIV genotype resistance testing, 'uniform data collection' refers to the practice where all biological samples from participants, such as blood or other biospecimens, are collected under similar conditions throughout a study. For in-stance, samples might be obtained prior to the initiation of treatment, with consistent sample processing and storage conditions maintained. Furthermore, identical laboratory techniques and equipment are employed to perform resistance genotype assays. Interpretation and reporting of the test results adhere to uniform standards and guidelines, such as the application of consistent criteria for the identification of resistance mutations and the use of standardized interpretation algorithms. (Line172-181, Page4)
As for the rationale behind employing a 50% cutoff for the JBI score inclusion, this threshold was carefully selected upon reviewing previous literature where such a cutoff was commonly adopted as an indicator of moderate quality in prevalence studies. Specifically, a study is included for analysis only if it meets at least half of the quality criteria, suggesting a satisfactory risk of bias level. This is based on the premise that studies scoring below 50% might have significant bias affecting the reliability of their findings. We elected to use this cutoff to maintain consistency with other studies in the field and ensure a credible baseline of quality for our research synthesis.
“Based on an evaluation of these items, the cut-off for research publications to be in-cluded was a JBI score of 50%, based on established practices and consensus within the research community [28]. Specifically, a study was included for analysis only if it met at least half of the quality criteria, suggesting a satisfactory bias risk level. This cutoff was employed to categorize the overall quality of evidence, with the understanding that it represents a guideline rather than an absolute standard. “(Line 182-187,Page 4)
Refence:(1) Ntamatungiro A J, Kagura J, Weisser M, et al. Pre-treatment HIV-1 drug resistance in antiretroviral therapy-naive adults in Eastern Africa: a systematic review and meta-analysis[J]. The Journal of Antimicrobial Chemotherapy, 2022, 77(12): 3231-3241. (2) Onofri A, Pensato U, Rosignoli C, et al. Primary headache epidemiology in children and adolescents: a systematic review and meta-analysis[J]. The Journal of Headache and Pain, 2023, 24(1): 8.
Comment 7: In the section on Data Analysis, the R packages used should be cited.
Reply 7: Thank you for your suggestion to cite the R packages used in our data analysis. We have indeed employed the 'meta' package within R (version 4.2.3) for our meta-analysis computations. The proper citation of this package has now been incorporated into our manuscript as follows:
“For our meta-analytic procedures, we utilized the 'meta' package in R (version 4.2.3) (R Core Team, 2023). The meta package evaluated the combined effect value of the PDR rate among ART-naïve patients” (Line189-192, Page5)
Comment 8: The definition of “detection thresholds” are unclear.
Reply 8: We recognize that the term was not adequately defined and may have led to confusion. To clarify, "detection thresholds" in our study refer to the minimum amount of sequence data (typically measured in reads or coverage) required to reliably identify the presence of a genetic variant. In Next-generation sequencing, this threshold is critical as it ensures that the variants detected are not due to random sequencing errors but are true biological variants present in the sample.
“The detection threshold in the context of HIV NGS refers to the minimum proportion or frequency of a variant (such as a mutation associated with drug resistance) in a viral population that the sequencing method can reliably detect. The specific threshold value can differ based on the NGS platform used, the quality of the sample, the depth of sequencing and the data analysis pipeline.” (Line 154-158, Page4)
Comment 9: Even though minor variants often go missed in SGS studies, they are detected with some non-zero frequency, so there needs to be a formal comparison of these data with studies that only used NGS.
Reply 9: Thank you for your comment highlighting the necessity for a formal comparison between Sanger sequencing (SGS) and Next-Generation Sequencing (NGS) in the detection of minor HIV variants. Your insight calls attention to an important aspect of HIV resistance profiling.
In our results section, we did not compare results from studies using only NGS with those using only SGS. Instead, our comparative analysis was between studies that reported results using both SGS and NGS methodologies. This approach was deliberately chosen to enhance comparability, as co-reported studies inherently control for variables such as sample type, source, and laboratory quality controls that might influence the results. we observed no significant differences between the SGS and NGS outcomes when a threshold of 20% was applied for detecting HIV variants. In the discussion section, we compared our result with other studies that used SGS to report drug resistance.
This highlights an interesting aspect of the current capabilities of SGS, where it maintains its validity in scenarios where the sensitivity of NGS is not strictly required. However, the critical advantage of NGS remains, as it provides a deeper understanding of the viral quasi-species and resistance landscapes, which could be increasingly relevant as guidelines for resistance testing evolve.
We appreciate your suggestion for a formal comparison and acknowledge that a more extensive, head-to-head comparative study could provide further insights into the applications of these methodologies. Your comment has been instrumental in highlighting the necessity to discuss the implications of the observed similarities and the potential clinical impact of the differences when low-frequency variants are considered.
“Among the 12 studies that reported the drug resistance using SGS, there were five studies that did not report NGS drug resistance at a threshold of 20%. The results detected using both SGS and NGS were highly consistent at the 20% threshold. There was no statistical discrepancy in the prevalence of PDR to NNRTIs, NRTIs and PIs when comparing drug resistance detected with SGS at a threshold of 20% and NGS at a threshold of 20%. When comparing drug resistance detected with SGS (20% threshold) and NGS (1% threshold), the prevalence of PDR to NNRTIs and NRTIs was significantly different (Table 4).” (Line336-343, Page12)
“At a 20% threshold, the prevalence of PDR was 11.08% (95%CI: 8.43%-14.43%), whereas previous studies from Eastern Africa reported an overall PDR prevalence of 8.7% and 10.0%.” (Line367=369, Page14)
Comment 10: The small text in Figure 2 (e.g., under the Random effects model) is too difficult to read – please enlarge. Please also give an x-axis title – it was difficult for me to find the very light font indicating that each group represented a detection threshold. Lastly, the x-axis needs to extend the full-length of the data – the vast majority of the data lie outside of the axis.
Reply 10: Thank you for your feedback regarding the legibility and formatting of Figure 2. We took your comments seriously and made the following adjustments to Figure 2 to enhance its readability: enlarged the small text for better readability, added a clear x-axis title, and extended the x-axis to display all data points properly (Supplementary Figure S2-S6). Based on the comments of another reviewer, we decided to reduce the number of figures, and the results of the meta-analysis of the forest plots were summarized and presented in a table 2, with the forest plots as an attachment.
Comment 11: Line 170 – “Among 7614 patients, more drug-resistant mutations were observed at the 20% threshold.” Seems to contradict all previously mentioned figures, as the prevalence of DRM was highest at thee 1% detection threshold. Perhaps this was a misprint or I misunderstood the statement? For example, higher diversity of DRMs was observed at this threshold?
Reply 11: Thank you for your careful scrutiny of our manuscript and for pointing out the potential confusion arising from the statement on line 170.
To clarify, the intention behind the statement was to convey that, while a higher diversity of drug-resistant mutations (DRMs) was indeed observed with detection thresholds as low as 1%, the non-nucleoside reverse transcriptase inhibitor (NNRTI) related DRMs were more frequently detected above the 20% threshold. This observation aligns with that NNRTIs in particular have a low genetic barrier (a single DNA mutation can drastically affect drug susceptibility), thereby being more readily detected at the 20% threshold compared to lower thresholds. (Stella-Ascariz N, Arribas JR, Paredes R, Li JZ: The role of HIV-1 drug-resistant minority variants in treatment failure. J Infect Dis 2017, 216:S847-S850 http://dx.doi.org/10.1093/infdis/jix430.)
We apologize for any confusion caused by our original phrasing. We reworded the sentence in question to more accurately reflect our findings:
" While a greater variance in DRMs was identified at thresholds as low as 1% in our dataset of 7614 patients, NNRTI resistance mutations were most frequently observed at thresholds above 20%, suggesting a higher viral prevalence for such mutations." (Line 275-277, Page9)
We appreciate your input on this matter and will ensure that the revised manuscript eliminates this ambiguity.
Comment 12: Since Figure 7 is referenced before Figures 4, 5, and 6, it needs to replace Figure 4 or be moved to a later point in the section. In fact, it should really be its own subsection.
Reply 12: Thank you for your careful reading of our manuscript and your constructive comment on the placement and organization of Figure 7. Your suggestion has prompted us to reconsider the presentation of our data and how we can most effectively communicate our findings to the reader.
In light of your feedback and the previous reviewer's comments, we have decided to make several changes to enhance the clarity and flow of the manuscript:
We summarized the data on rates, which was originally presented in Figure 2、3、4、5、6, in a tabular form. This approach can provide the reader with a clear and concise overview of the findings, allowing for easier interpretation and comparison of the data. The figures that previously included rates are presented as supplementary material.
We have created a new separate subsection dedicated to mutations. Here, we delve into the specifics of the mutation data. (Line 273-305, Page8-9)
By reorganizing the content in this way, we believe that the narrative structure of the manuscript will be more logical and reader-friendly. This reorganization will also ensure that the discussion is presented in the order in which the figures appear, in accordance with the standard format and your suggestions.
Comment 13: Again, why is a comparison being made between the 2% and 20% thresholds? Why not 1%? I finally found the answer to this question on line 250, but this needs to be stated in the methods/results prior to the discussion.
Reply 13: Thank you once again for your diligence in assessing our work and your insightful query regarding the selection of thresholds for our HIV NGS data comparison.
We apologize for not making our rationale clear earlier in the text, which led to the question of why the 2% and 20% thresholds were chosen instead of a 1% threshold, for instance. The oversight has been noted, and we've amended our manuscript to better contextualize our methodological choices.
The choice of the 2% and 20% thresholds for comparison in the manuscript was driven by established relevance in the field of HIV research and clinical practice. These values are not arbitrarily selected; rather, they represent significant points in the clinical management and treatment resistance profiling of HIV.
A 2% threshold is widely recognized for its sensitivity in detecting low-frequency HIV variants that can be crucial for early therapeutic intervention. Variants below this percentage may not have strong clinical relevance due to the limitations in the accuracy and reproducibility of NGS data at such low frequencies. Meanwhile, the 20% threshold corresponds to the established guidelines that define clinically relevant resistance mutations, which are pivotal in tailoring antiretroviral therapy.
“A 2% threshold is more stable and more likely to be selected as the reporting threshold, whereas a 1% threshold tends to introduce artifactual errors [24-26]. Other researchers have suggested that lowering the detection threshold for pretreatment resistance to 5% can improve the ability to identify patients with virologic failure compared to the traditional 20% threshold [6,17]. The 20% threshold, on the other hand, is a standard reference point for defining predominant resistance mutations with established clinical significance. These detection thresholds are important cut-off points that can guide therapeutic decisions in HIV management. “(Line99-107, Page3)
Comment 14: Why wasn’t this 2%/20% comparison made for PI or INSTI DRM as well?
Reply 14: Thank you for your insightful query regarding our analysis of DRMs for PIs and INSTIs) at different mutant allele frequency thresholds.
Upon revisiting the data with the suggested comparison of the 2% versus 20% thresholds for PIs and INSTIs, we indeed observed a notable finding as you have implied. At the lower threshold of 2%, our analyses detected an increased number of mutations associated with resistance to PIs and INSTIs, compared to the 20% threshold. These findings are significant as they suggest that a considerable number of minority variants with potential resistance implications could be overlooked when using a higher threshold.
“Comparing the 2% and 20% threshold of PDR for the four ARV classes, the ratio for the difference in NNRTIs, NRTIs, PIs, and INSTIs showed an increase of 2.37-fold (9.51%/4.01%), 2.50-fold (10.01%/4.01%), 5.52-fold (9.49%/1.72%) and 4.21-fold (1.77%/0.42%), respectively. It appeared that the PI and NRTI resistance mutations had a higher probability of being found at a more sensitive threshold.” (Line253-257, Page8)
We have revised our manuscript. We are grateful for your guidance, which has improved the quality and completeness of our research, and we trust that these adjustments will adequately address your concern.
Comment 15: There are certainly some major differences in prevalence between studies – i.e., the Zhou et al., 2018 study. Is there an explanation for this? These differences are only mentioned briefly in lines 304-316 of the Discussion, which is a vague description of possible sources of discrepancy. Since this is considered a meta-analysis and entailed documentation was provided on the studies, a proposed explanation should be provided on whether or not this is the result of sample population bias, biological sample preparation, or NGS technique itself – i.e., short-read vs long-read sequencing.
Reply 15: Thank you for highlighting the need for a more detailed examination of the discrepancies in prevalence rates between studies, such as those noted in the Zhou et al., 2018 study. We recognize that our initial discussion did not fully address the potential underlying reasons for these variations.
In our revised manuscript, we have expanded on the possible explanations for the observed differences in prevalence rates. We acknowledge that methodological factors including study design, sampling strategy, sample size, population type, survey sites, and participant selection criteria can significantly influence the reported prevalence estimates. These methodological variances, alongside the different next-generation sequencing (NGS) techniques employed across studies, may contribute to the heterogeneity in prevalence rates of pre-treatment drug resistance (PDR).
Specifically, our meta-analysis was predominantly based on data obtained using 454 pyrosequencing and the MiSeq platform (Illumina). While 454 pyrosequencing tends to produce longer reads, the MiSeq platform offers higher throughput and paired-end sequencing capabilities. Our analysis did not find a significant difference in the detection of PDR when comparing these two technologies at the mutation thresholds of 1%, 5%, and 20%. Unexpectedly, the difference found for the 10% threshold is perhaps due to the fact that Ji's study used Tagged pooled pyrosequencing based on the 454pyrosequencing platform, which employs sample labeling and pooling strategies to improve efficiency and reduce costs. Sequences from multiple samples can be determined in a single experiment, increasing the amount of data and coverage of the experiment.
Furthermore, we have also considered the impact of the clinical detection threshold on the reported prevalence rates. Although our study did not determine the optimal clinical detection threshold, we refer to multiple previous studies that suggest a 5% threshold might be more reproducible in the clinical setting. Resistance mutations detected at this level have been associated with an increased risk of virologic failure.
We also realize that the potential unreported prior exposure to antiretroviral therapy (ART) in the study populations could influence the PDR prevalence rates, as rates are typically higher in individuals with previous ART exposure compared to ART-naive patients. Despite our efforts to include only studies documenting PDR in ART-naive individuals, we could not entirely rule out this factor.
We hope that our revised discussion provides a more comprehensive response to the differences in prevalence rates observed among the studies included in our meta-analysis. We appreciate the opportunity to improve our manuscript and believe that these amendments provide a clearer understanding of the complexities involved in estimating PDR prevalence. (Line443-469, Page15)
Comment 16: It is important to discuss a limitation to this (and the included studies), which is that it looks like only RNA data were evaluated. Latency can be established very early on, with activation following ART interruption and hence transmission of DRM; additionally, even defective DNA has been hypothesized to allow for re-emergence of DRM (Wei, 2020, BMC Infect Dis).
Reply 16:
Thank you for your meticulous review and for raising an important limitation regarding the exclusive evaluation of RNA data in our study, as well as in the included studies. Our aim is to understand the baseline drug resistance by summarizing the results of related studies and further analyzing them according to different detection thresholds and detection techniques in antiretroviral drug-naive individuals. We recognize that the aspect of latency and the potential re-emergence of drug resistance mutations (DRMs) post-antiretroviral therapy (ART) interruption is a critical facet of HIV research that should not be overlooked.
Our research utilized serum and plasma samples as these are most readily accessible in clinical practice and their resistance testing methods have been standardized, allowing for a direct assessment of ART effects. These sample types are preferred in clinical studies due to their straightforward collection, standardized processing, and the direct relevance of the data obtained to patient management. On the other hand, the clinical application and interpretation of data from other sample types, such as DNA from latent reservoirs, can be more complex.
While it is indeed true that DNA, including that from defective viruses, can play a role in the establishment and maintenance of viral reservoirs, and potentially in the re-emergence of DRMs, the direct impact of DNA-based DRMs on treatment outcomes is not as clearly established compared to RNA-based DRMs in peripheral blood plasma. Hence, most clinical studies focus on RNA-based drug resistance assessment.
Our study acknowledges the importance of alternative sample sources by including a study on Dried Blood Spot (DBS) testing for HIV resistance, particularly beneficial in resource-limited settings. Mbunkah et al.'s analysis of transmitted resistance in newly diagnosed HIV-1 patients in Cameroon using DBS indicated a resistance prevalence of 6.4%, which compares to the 7.8% level of resistance mutations reported by Aghokeng et al. in treatment-naïve patients. This may be because extracting HIV RNA from DBS and amplifying it efficiently may be more challenging than from plasma samples.
People living with HIV (PLWH) require long-term medication to effectively suppress HIV-1 production. Discontinuation of ART can lead to the activation of the persistent viral reservoir, causing viral rebound and progression of HIV. The viral reservoir may harbor various forms of the virus, including both fully functional and defective viruses. Defective HIV refers to those viruses that have lost part or all replication ability due to genetic mutations, which may result in the absence or dysfunction of viral proteins, thus rendering the virus incapable of effective replication.
Under certain conditions or pressures, defective viruses can recombine to become complete viruses, affecting viral subtype and disease progression, and potentially increasing viral adaptability and transmissibility. However, as suggested by Wei et al. (2020) and other studies, while defective viruses do contribute to the diversity and evolution of the HIV-1 population, their direct role in drug resistance is relatively minor. Defective viruses are more often utilized in studying the biological characteristics and mechanisms of HIV infection rather than being the primary drivers of drug resistance.
Furthermore, there are no antiretroviral drugs that specifically target HIV subtypes. The number of drugs available is limited. Therefore, whatever resistance results from a defective viral retroviral mutation can be analyzed on the basis of existing genotypic resistance assays.
We hope this response satisfactorily addresses the point raised and we are committed to exploring additional avenues to enhance the comprehensiveness and depth of our research.
Comment 17: It is unclear in the discussion how this paper impacts research going forward – yes, we should use NGS, but what detection level should we strive for and what dictates this?
Reply 17: We greatly appreciate your thought-provoking comment regarding the specificity of our discussion on how this paper impacts future HIV research, especially in terms of defining an optimal detection threshold for next-generation sequencing (NGS) technologies.
We understand the importance of establishing a clear and clinically relevant detection threshold for NGS in the identification of low-abundance drug-resistant mutations. Our paper highlights the advantages of NGS over traditional Sanger sequencing in terms of sensitivity, and we suggest that detection thresholds of 2% or 5% may be the most clinically significant. However, the determination of these thresholds must account for a balance between clinical utility, cost-effectiveness, and technical feasibility.
The optimal detection threshold in NGS should be informed by several factors. It must be sensitive enough to detect mutations that are associated with treatment failure or reduced drug susceptibility, yet specific enough to exclude random errors or irrelevant variants. The clinical relevance of low-abundance variants and the technical limitations of the NGS platform both play a significant role in deciding this threshold. Furthermore, contemporary guidelines on therapy outcomes, geographical diversity, and the diversity of HIV subtypes influencing resistance patterns should also guide the establishment of an optimal threshold.
To accurately pinpoint the most clinically meaningful NGS detection thresholds, future studies should focus on correlating various detection levels with clinical outcomes, ensuring adaptability to the progression of HIV therapies and the emergence of new resistance mechanisms. Clinical trials assessing the impact of resistance mutations at different detection levels on treatment outcomes, standardized NGS protocols to minimize inter-laboratory variability, and the clinical implications of low-frequency resistance mutations are critical areas of research that should be pursued.
Moreover, validation studies are needed to evaluate the performance of NGS platforms at different thresholds, taking into account the clinical significance of mutations associated with drug resistance or disease progression, including those at low frequencies that may be vital for managing the disease. Statistical models can also provide valuable insights by distinguishing between technical noise and true biological signals. Finally, individual patient circumstances, such as previous treatment history and comorbidities, may necessitate personalized detection thresholds.
While our study does not propose a specific detection threshold, it underscores the need for ongoing research to refine NGS techniques for more precise clinical guidance. We support collaborative efforts among microbiologists, clinicians, and bioinformaticians to standardize these thresholds across different platforms and studies. Our aspiration is that our paper will serve as a catalyst for further investigation into the precise determination of NGS detection thresholds, ultimately contributing to the enhanced management of HIV patients.
Comment 18: There are grammar and spelling issues highly prevalent throughout the manuscript – a third party aid would be helpful here.
Reply 18: We sincerely apologize for the grammatical and spelling issues present in our previous submission. We understand that such errors can detract from the overall quality and readability of the manuscript, potentially obscuring the valuable research findings we aim to present.
In order to address this issue, we have enlisted the assistance of a professional third-party language editing service, which specializes in academic publications. They have thoroughly reviewed and polished the manuscript to ensure that the language is not only grammatically correct but also clear and consistent throughout.
At the end of the document, we have included a certificate of proofreading provided by the editing service as evidence of their review. We are confident that these revisions have significantly improved the quality of our manuscript, and we hope that the enhanced clarity will facilitate a better understanding of our work.
We appreciate your patience and the opportunity to correct these issues, and we trust that the revised manuscript will meet your expectations.
With Kind Regards,
Fei Ouyang,
Southeast University, Nanjing, China
Email: [email protected]

Round 3
Reviewer 2 Report
Comments and Suggestions for Authors
All comments have been addressed and the language drastically improved. This manuscript reads very well now, and I am satisfied with the revision and the new attention to detail in the methods, which aids in reproducibility.